# Uncertainty-bounded estimates of ash cloud properties using the ORAC algorithm: Application to the 2019 Raikoke eruption

Andrew T. Prata[1], Roy G. Grainger[1], Isabelle A. Taylor[2], Adam C. Povey[3], Simon R. Proud[3, 4], and Caroline A. Poulsen[5]

[1]Atmospheric, Oceanic and Planetary Physics, University of Oxford, Oxford OX1 3PU, UK
[2]COMET, Atmospheric, Oceanic and Planetary Physics, University of Oxford, Oxford OX1 3PU, UK
[3]National Centre for Earth Observation, Atmospheric, Oceanic and Planetary Physics, University of Oxford, Oxford OX1 3PU, UK
[4]RAL Space, STFC Rutherford Appleton Laboratory, Harwell Campus, Didcot OX11 0QX, UK
[5]Australian Bureau of Meteorology, Melbourne, Australia

**Correspondence:** Andrew Prata (andrew.prata@physics.ox.ac.uk)

**Abstract.** Uncertainty-bounded satellite retrievals of volcanic ash cloud properties such as ash cloud-top height, effective radius, optical depth and mass loading are needed for the robust quantitative assessment required to warn aviation of potential hazards. Moreover, there is an imperative to improve quantitative ash cloud estimation due to the planned move towards quantitative ash concentration forecasts by the Volcanic Ash Advisory Centers. Here we apply the Optimal Retrieval of Aerosol and Cloud (ORAC) algorithm to Advanced Himawari Imager (AHI) measurements of the ash clouds produced by the June 2019 Raikoke (Russia) eruption. The ORAC algorithm uses optimal estimation to consolidate *a priori* information, satellite measurements and associated uncertainties into uncertainty-bounded estimates of the desired state variables. Using ORAC, we demonstrate several improvements in thermal infrared volcanic ash retrievals applied to broadband imagers. These include: an improved treatment of measurement noise, accounting for multi-layer cloud scenarios, distinguishing between heights in the troposphere and stratosphere, and the retrieval of a wider range of effective radii sizes than existing techniques by exploiting information from the 10.4 µm channel. Our results indicate that $0.73 \pm 0.40$ Tg of very fine ash (radius $\leq 15$ µm) was injected into the atmosphere during the main eruptive period from 21 June 18:00 UTC to 22 June 10:00 UTC. The total mass of very fine ash decreased from 0.73 Tg to 0.10 Tg over $\sim$48 h with an $e$-folding time of 20 h. We estimate a distal fine ash mass fraction of $0.73 \pm 0.62$ % based on the total mass of very fine ash retrieved and the ORAC-derived height time-series. Several distinct ash layers were revealed by the ORAC height retrievals. Generally, ash in the troposphere was composed of larger particles than ash present in the stratosphere. We also find that median ash cloud concentrations fall below peak ash concentration safety limits ($<4$ mg m$^{-3}$) 11–16 h after the eruption begins, if typical ash cloud geometric thicknesses are assumed. The ORAC height retrievals for the near-source plume showed good agreement with GOES-17 side-view height data ($R = 0.84$, bias = -0.75 km); however, a larger negative bias was found when comparing ORAC height retrievals for distal ash clouds against Cloud-Aerosol Lidar with Orthoganol Polarisation (CALIOP) measurements ($R = 0.67$, bias = -2.67 km). The dataset generated here provides uncertainties at the pixel level for all retrieved variables and could potentially be used for dispersion model validation or implemented in data assimilation schemes. Future work should focus on improving ash

detection, improving height estimation in the stratosphere and exploring the added benefit of visible channels for retrieving effective radius and optical depth in opaque regions of nascent ash plumes.

## 1 Introduction

Volcanic Ash Advisory Centers (VAACs) require satellite observations to detect and track volcanic ash clouds that pose a threat to aviation. In addition to detection and tracking, VAACs use dispersion models to forecast the position of a volcanic ash cloud. Currently, VAACs are required to provide qualitative, deterministic forecasts indicating the future position of potentially hazardous ash clouds and satellite detection schemes have been developed in support of this operational requirement (Pavolonis et al., 2015a, b). However, according to the International Airways Volcano Watch (IAVW) roadmap, by 2025[1], VAACs will be required to issue quantitative forecasts of ash concentration (ICAO, 2019). The move to quantitative dispersion model forecasts motivates the need for validation to improve the overall quality of the forecast. The thermal infrared (IR) capabilities of the current generation of geostationary satellite sensors are particularly well suited to this purpose as they offer continuous, day and night observations at finer spatial (2 km) and temporal (10 minute) resolutions than operational dispersion model output grids (typically 10–25 km and 1–3 h). Thermal IR geostationary satellite observations are useful for quantitative validation (Wilkins et al., 2016; Prata et al., 2021; Folch et al., 2022), source term characterisation (Pouget et al., 2013; Van Eaton et al., 2016; Prata et al., 2020), data insertion (Wilkins et al., 2015, 2016; Folch et al., 2020; Prata et al., 2021), data assimilation (Lu et al., 2016; Pardini et al., 2020; Zidikheri and Lucas, 2021; Mingari et al., 2022) and source term inversion (Stohl et al., 2011; Harvey et al., 2020, 2022). In addition, it is important that uncertainties in satellite retrievals are accurately characterised because the VAACs and other users of satellite retrievals increasingly require uncertainty information to correctly interpret, aggregate and utilise the data.

Volcanic ash retrievals from satellite data are possible in the thermal IR because of the high $SiO_2$ content of volcanic ash, which has a strong absorption feature around the 9.5 μm wavelength region (Soda, 1961; Grainger et al., 2013; Prata et al., 2019). Volcanic ash clouds can be discriminated from water and ice clouds because the absorption of thermal radiation for ash decreases from 10–12 μm while thermal infrared absorption increases from 10–12 μm for water and ice, a property known as the 'reverse absorption' effect (Prata, 1989a, b). Wen and Rose (1994) and Prata and Grant (2001) demonstrated how estimates of the effective radius and optical depth (at 11 μm) could be obtained on the basis of two split-window brightness temperature measurements centred near 11 and 12 μm. Based on this principle, mass loadings (mass per unit area) can be derived at pixel-scale resolution and when combined with vertical profile information, ash concentrations can be estimated (Prata and Prata, 2012).

Uncertainties in satellite-based ash cloud retrieval algorithms are dominated by inaccuracies in the physical model whereby ash cloud properties or 'state variables' (such as cloud-top pressure or height, optical depth and effective radius) are converted into satellite-measured radiances. This is achieved using a radiative transfer forward model (FM) which simulates top-of-the-atmosphere (TOA) radiances based on certain assumptions about the atmospheric state. Numerous simplifying assumptions

---

[1]2026/2027 for the full requirement to be made (C. Lucas 2022, pers. comm.)

are needed to evaluate the FM and each of these assumptions introduce FM uncertainty. Examples include: parameterising a cloud layer in terms of its microphysical, optical and geometric properties; assuming whether or not it is plane-parallel (i.e. neglecting 3D radiative effects); parameterising the underlying surface characteristics. Additional uncertainties arise due to uncertainty in ancillary information, often derived from reanalysis data derived from Numerical Weather Prediction (NWP) models, such as profiles of temperature, humidity and trace gases. Uncertainties related to the measurements must also be

considered, which include channel noise and co-registration as well as scene-dependent uncertainties introduced by sub-pixel scale inhomogeneity.

Uncertainty in the total mass of very fine ash (radius $\leq$ 15 μm) derived using the split-window method has been estimated from sensitivity analyses and is generally found to be 40–60% (Wen and Rose, 1994; Gu et al., 2003; Corradini et al., 2008; Prata and Prata, 2012; Prata et al., 2017b). Sources of FM uncertainty that have previously been considered include the assumed

ash composition (e.g. Wen and Rose, 1994; Mackie et al., 2014; Prata et al., 2019; Deguine et al., 2020; Piontek et al., 2021b), form of the underlying size distribution (e.g. Wen and Rose, 1994; Western et al., 2015), variation in surface and cloud-top temperature (e.g. Corradini et al., 2008; Schneider et al., 1999), particle shape (Kylling et al., 2014) and meteorological cloud interference (Kylling et al., 2015). While broad estimates of uncertainty in the total mass of very fine ash are useful, it is desirable to have pixel-scale, uncertainty-bounded estimates of all retrieved quantities.

Optimal estimation (OE; Rodgers, 1976, 2000) is particularly useful for solving the noisy inverse problem where imperfect prior knowledge in the state space is quantified in terms of probability density functions (PDFs) and noise in the measurements is quantified in terms of PDFs in the measurement space. Within the OE framework, one can assume large uncertainties on the priors such that the solution is influenced mainly by the measurements or set small uncertainties if good quality *a priori* information is available. Optimal estimation also provides a formalism where state and measurement variables can be easily

added or removed and uncertainties in prior information and the measurements are propagated through the FM equations to estimate uncertainties in all retrieved state variables.

There are several examples of OE-retrievals that use thermal-only measurements of volcanic ash, each using different state variables and instrument channels but all aiming to retrieve the ash mass loading and cloud-top height. For example, the Pavolonis et al. (2013) algorithm, used by the Washington, Anchorage and Darwin VAACs, includes the effective cloud temperature,

effective cloud emissivity (at 11 μm) and the '$\beta$-ratio' (the ratio of effective optical depth at 12 and 11 μm) in the state vector. From these state variables, cloud-top height and mass loading are derived. The measurement vector used by Pavolonis et al. (2013) includes the 11 μm brightness temperature and two brightness temperature differences (BTDs; 11-12 μm and 11-13.3 μm). Uncertainties considered by Pavolonis et al. (2013) include measurement (instrument) error, clear-sky radiance errors (land and water are treated separately) and spatial heterogeneity errors. For the measurement errors in each channel,

a fixed, noise-equivalent delta temperature (NE$\Delta$T), as reported by the satellite provider, is used. The Francis et al. (2012) algorithm, used by the London VAAC, includes the effective radius, cloud-top pressure and mass loading in their state vector and use the 11, 12 and 13.3 μm brightness temperatures in their measurement vector. Similar to Pavolonis et al. (2013), Francis et al. (2012) also use fixed NE$\Delta$T values to estimate measurement uncertainty for each channel included in the measurement vector while FM uncertainty is estimated using statistics derived from long-term, cloud-free satellite radiances. Kylling et al.

(2015) present a simple OE scheme that retrieves two state variables (effective radius and optical depth at 11 µm) based on two measurements (11 and 12 µm brightness temperatures) from which mass loading is computed. Cloud-top height is not provided in their paper, but could have been inferred from their estimation of cloud-top temperature using a nearby meteorological temperature profile (from a NWP model or sounding). To characterise uncertainty in their retrieval, Kylling et al. (2015) use the combined (FM and measurement) uncertainties provided in Francis et al. (2012).

A common theme amongst existing retrieval schemes is the use of a fixed measurement error (i.e. NEΔT) per channel. However, this noise estimate is only true for the reference temperature given. It is straightforward to allow NEΔT to vary based on the measured brightness temperature and we highlight this improvement in the present study (Sect. 3.2.2). Another key result from the work of Pavolonis et al. (2013) and Francis et al. (2012) is that the inclusion of the 13.3 µm channel enables good estimates of ash cloud-top height, as shown by validation with the Cloud-Aerosol Lidar and Infrared Pathfinder Satellite

Observation (CALIPSO) satellite lidar measurements presented in their studies. As we will show, the 13.3 µm channel is key for distinguishing between ash clouds in the troposphere and stratosphere. Additionally, no authors have yet published an ash OE-scheme that incorporates the 10.4 µm channel in addition to channels centred near 11, 12 and 13.3 µm. The 10.4 µm channel is a window channel less affected by water vapour than the 11 µm and 12 µm channels (Lindsey et al., 2012) and is positioned closer to the 9.5 µm silica absorption band.

We focus on quantifying ash cloud properties and their associated uncertainties using the Optimal Retrieval of Aerosol and Cloud (ORAC) algorithm applied to Advanced Himawari Imager (AHI) observations using the June 2019 Raikoke (Russia) eruption as a case study. The main aim of this study is to provide uncertainty-bounded estimates of optical depth (at 550 nm), effective radius, cloud-top height and mass loading for the Raikoke ash clouds, while simultaneously describing new advances in thermal-only volcanic ash retrievals applied to satellite imager instruments using an OE framework. Specifically, we present

a time-series of the Raikoke ash cloud properties and discuss our results in the context of existing studies on Raikoke. We present a new method for quantifying ash cloud heights in the troposphere and stratosphere (a common feature of explosive volcanic eruptions such as Raikoke). We also present the first ash retrievals for the Raikoke case which consider multi-layered cloud (ash over water cloud) and present a technique for selecting the best-fitting FM based on measurement cost at solution. We consider uncertainty variation in measured brightness temperatures which advances previous methods of assuming constant

brightness temperature errors. Finally, we explore the advantages of using the 10.4 µm channel in OE retrievals of volcanic ash and demonstrate that its inclusion enables the retrieval of a wider range of effective radii sizes.

## 2 Data

### 2.1 Advanced Himawari Imager

The Advanced Himawari Imager (AHI) aboard the Japanese Meteorological Agency's (JMA) Himawari-8 satellite has been in

operation since 7 July 2015. The AHI is in geostationary orbit nominally positioned at 140.7° E and completes a full disk scan every ten minutes (Bessho et al., 2016). The AHI has sixteen spectral bands with spatial resolutions at nadir of 0.5 km (band 3; 0.64 µm), 1 km (bands 1, 2 and 4; 0.47, 0.51 and 0.86 µm) and 2 km (bands 5–16; 1.6, 2.3, 3.9, 6.2, 6.9, 7.3, 8.6, 9.6, 10.4, 11.2,

12.4 and 13.3 µm). The ORAC algorithm has been applied to level 1b Himawari Standard Data (HSD) files, which contain radiances that have been sampled onto the World Geodetic System 1984 (WGS84) ellipsoid. The radiances were converted to

brightness temperatures using the JMA's standard calibration (calibration information block 5 of the header). A subset of the AHI full disk defined by 135° E–15° W and 40–65° N has been analysed from 18:00 UTC on 21 June 2019 to 18:00 UTC on 28 June 2019. This corresponds to one week of observations from eruption onset.

## 2.2 Advanced Baseline Imager

To validate ORAC-AHI retrievals of cloud-top height in the near-source volcanic plume we use the recently published dataset

from Horváth et al. (2021a, b) who provide geometric estimations of plume-top height. Specifically, we use the GOES-17 Advanced Baseline Imager (ABI; positioned at 137.2° W) 'side-view' height estimations provided in the Supplementary Material in Horváth et al. (2021b). Uncertainty on these height retrievals is $\pm$ 500 m.

## 2.3 CALIOP

To validate the ORAC-AHI retrievals of cloud-top height for distal ash clouds, we use the level 2 lidar products generated

from measurements made by the Cloud-Aerosol Lidar with Orthoganol Polarisation (CALIOP) aboard the CALIPSO platform (Winker et al., 2009). The level 2 version 4.20, 5 km combined cloud and aerosol layer product (L2_05kmMLay-Standard-V4-20) is used to extract ash cloud heights and geometric thicknesses for validation purposes. The precision with which CALIOP measures layer top and base height varies with altitude. From -0.5 to 8.2 km, the vertical resolution is 30 m and from 8.2 to 20.2 km, the vertical resolution is 60 m.

# 3 Method

## 3.1 Forward model

The ORAC algorithm is an open source software initially developed by the University of Oxford and Rutherford Appleton Laboratory (RAL). The Deutscher Wetterdienst (DWD) has developed the code alongside Oxford and RAL since 2010. The Australian Bureau of Meteorology is also a contributor. Within ORAC two distinct FM implementations are available to

retrieve aerosol or cloud properties (or combined for a joint retrieval). A cloud is considered as a geometrically infinitesimal layer within the atmosphere whereas an aerosol layer is considered as a continuum. Here we use the FM representing cloud because volcanic ash clouds are often observed as well-bounded features in the vertical (e.g. Prata et al., 2017a) rather than a well-mixed continuum distributed over a vertical region of the atmosphere. This choice has a practical advantage in that the code for the cloud FM is setup for day and night retrievals whereas the aerosol FM currently only permits daytime retrievals.

Details of the aerosol FM are provided in Thomas et al. (2009) and the cloud FM details can be found in Poulsen et al. (2012) and McGarragh et al. (2018). As we use the ORAC FM for cloud, no further details of the aerosol FM are given here. In the cloud model, the solar and thermal components of terrestrial radiation are considered separately, but are minimised

simultaneously. As the Raikoke ash clouds dispersed into the atmosphere over several days and nights, we only use thermal channels in our measurement vector to ensure consistency over day and night in the retrievals. As we only use thermal channels

in our measurement vector, we focus on the thermal FM within ORAC.

The top-of-the-atmosphere radiance ($L_{\mathrm{TOA}}$), measured by a downward-looking satellite, for a plane-parallel cloud at thermal wavelengths (3–15 μm) can be written as (McGarragh et al., 2018)

$$L_{\mathrm{TOA}} = L_{\mathrm{ac}}^{\uparrow} + [L_{\mathrm{ac}}^{\downarrow} R_{\mathrm{db}}^{\uparrow}(\theta_v) + B(T_{\mathrm{c}})\epsilon(\theta_v) + L_{\mathrm{bc}}^{\uparrow} t_{\mathrm{db}}^{\uparrow}(\theta_v)]t_{\mathrm{ac}}(\theta_v), \tag{1}$$

where $L_{\mathrm{ac}}^{\uparrow}$ is the above cloud upwelling radiance, $L_{\mathrm{ac}}^{\downarrow}$ is the above cloud downwelling radiance, $R_{\mathrm{db}}^{\uparrow}(\theta_v)$ is the portion of above

cloud, downwelling radiance reflected towards the viewing direction of the satellite, $\theta_v$, $B(T_{\mathrm{c}})$ is the Planck radiance at the cloud-top temperature, $T_{\mathrm{c}}$, $\epsilon(\theta_v)$ is the cloud emissivity, $L_{\mathrm{bc}}^{\uparrow}$ is the below cloud upwelling radiance, $t_{\mathrm{db}}^{\uparrow}(\theta_v)$ is the upward diffuse transmittance of the cloud and $t_{\mathrm{ac}}(\theta_v)$ is the above cloud transmittance of the atmosphere. Note that if we assume that the reflected portion of above cloud, downwelling thermal radiance is negligible (i.e. $L_{\mathrm{ac}}^{\downarrow} R_{\mathrm{db}}^{\uparrow} \approx 0$) and recognise that $t_{\mathrm{db}}^{\uparrow}(\theta_v) = 1 - \epsilon(\theta_v)$ then Eq. 1 reduces to the FM formulations used in Pavolonis et al. (2013) and Francis et al. (2012):

$$L_{\mathrm{TOA}} = \epsilon L_{\mathrm{cld}} + (1 - \epsilon)L_{\mathrm{clr}}, \tag{2}$$

where

$$L_{\mathrm{clr}} = L_{\mathrm{bc}}^{\uparrow} t_{\mathrm{ac}} + L_{\mathrm{ac}}^{\uparrow} \tag{3}$$

and

$$L_{\mathrm{cld}} = L_{\mathrm{ac}}^{\uparrow} + B(T_{\mathrm{c}})t_{\mathrm{ac}}. \tag{4}$$

Further, if it is assumed that the atmosphere is perfectly transparent and the surface has an emissivity of 1, then $L_{\mathrm{clr}} = B(T_s)$ and $L_{\mathrm{cld}} = B(T_{\mathrm{c}})$ and we arrive at the original formulation proposed by Prata (1989a) (i.e. $L_{\mathrm{TOA}} \approx \epsilon B(T_{\mathrm{c}}) + (1 - \epsilon)B(T_s)$). Therefore, although there are significant differences in the practical implementation of the ORAC thermal FM and those of Pavolonis et al. (2013) and Francis et al. (2012), the main difference in its theoretical formulation is the inclusion of non-zero above cloud reflectance of downwelling radiance (as written in Eq. 1). Another important difference is that we include surface

temperature, $T_s$, in the state vector. This means that $L_{\mathrm{bc}}^{\uparrow}$ must be updated during the retrieval process. For computational efficiency $L_{\mathrm{bc}}^{\uparrow}$ is written as a linear expansion in temperature (McGarragh et al., 2018):

$$L_{\mathrm{bc}}^{\uparrow} = L_{\mathrm{bc,a}}^{\uparrow} + (T_s - T_{\mathrm{s,a}})\frac{\partial B(T_{\mathrm{s,a}})}{\partial T_{\mathrm{s,a}}}\epsilon_s t_{\mathrm{bc}}(\theta_v), \tag{5}$$

where $T_{\mathrm{s,a}}$ is the *a priori* surface temperature (taken from NWP data), $\epsilon_s$ is the emissivity of the surface and $t_{\mathrm{bc}}(\theta_v)$ is the transmittance from the surface to the cloud layer. The clear-sky radiance and transmittance terms are computed using version

13 of RTTOV (Radiative Transfer for TOVS; Saunders et al., 2018). As a pre-processing task, RTTOV is run on atmospheric profiles of temperature, specific humidity and ozone taken from ERA5 reanalysis data (Hersbach et al., 2020). The ERA5

data are interpolated in time to match the satellite observation time from a 0.5° × 0.5° global grid at 6 h temporal resolution. Clear-sky, above cloud and below cloud transmittance and radiance profiles are then passed to ORAC. A cloud layer is inserted into the FM by interpolating the above and below radiance and transmittance profiles to the first guess pressure of the state vector. Since the cloud layer is assumed to be a geometrically (but not optically) infinitesimal layer, this implementation is fast and flexible because once the pre-processing task is done, any cloud optical properties can be introduced or modified. In addition, this approach allows for both single and multi-layer FM configurations and delegates the generation of computationally expensive single-scattering cloud properties to offline calculations.

The single-scattering properties for the cloud layer are generated as look-up tables (LUTs) using version 2.1 of DISORT (Stamnes et al., 2000). Volcanic ash LUTs are generated as a function of the 550 nm optical depth ($\tau$; 17 grid points from 0–256 with $\log_{10}$ spacing), effective radius ($r_e$; 13 grid points at 0.1 µm, 0.5 µm and 1–15 µm in 1 µm intervals), satellite zenith angle ($\theta_v$; 10 grid points from 0–90° in 10° intervals) and wavelength ($\lambda$; convolved to the relevant channel spectral response function). Ash composition information is accounted for using complex refractive index data taken from the Oxford Aerosol Refractive Index Archive (ARIA, http://eodg.atm.ox.ac.uk/ARIA/). As we do not have refractive index data for the Raikoke ash, we ran the retrieval with three different types of ash (with varying bulk silica content), reported by Reed et al. (2018), that were sampled from the 2010 Eyjafjallajökull (Iceland) eruption, 1981 Mt Spurr (Alaska, USA) eruption and 2008 Chaitén (Chile) eruption. After running the ORAC retrieval for all three ash compositions, we found that the Eyjafjallajökull ash consistently outperformed the other two ash compositions (i.e. lower cost and more retrievals converging). The bulk silica contents for Eyjafjallajökull, Mt Spurr and Chaitén are 58.85, 55.99 and 74.90 wt% (see Prata et al., 2019, Table 2). Smirnov et al. (2021) provide bulk silica contents for samples representing the 21–26 June 2019 eruption and show that for "glass compositions of shards from air fall ash" bulk silica contents are mostly between 57–63 wt% (see their Total Alkali Silica diagram in Fig. 5 top panel). Therefore the bulk composition for the Eyjafjallajökull ash (58.85 wt%) appears to be consistent with the glass shards of air fall ash reported by Smirnov et al. (2021). We therefore present retrieval results only for the Eyjafjallajökull ash composition. Particles were assumed to be spherical and the underlying size distribution was assumed to follow a lognormal distribution with a spread of 2. While ash particles are known to be of irregular shape, here we assume they are spherical for two main reasons: (1) At thermal infrared wavelengths larger than 10 µm, uncertainty due to unknown particle habit (non-sphericity) is expected to have little impact on the retrievals as discussed by numerous previous authors (Wen and Rose, 1994; Corradini et al., 2008; Clarisse et al., 2010; Newman et al., 2012; Pavolonis et al., 2013; Prata et al., 2017b). Yang et al. (2007) show that the impact of non-sphericity at thermal infrared wavelengths is negligible for desert dust (which is similar in many ways to volcanic ash). (2) The exact non-spherical shapes of the Raikoke ash particles under investigation here are unknown. Therefore, approximating their shape by some other irregular shape may introduce further error than simply assuming a sphere. We recognise that it's possible to find differences between spherical and non-spherical particles if irregular, porous objects are compared with spheres. Kylling et al. (2014) found that differences in the total mass uncertainty would increase from 40% to 50%. However, it is questionable how representative the particle shapes used in the Kylling study are for the Raikoke ash and therefore we cannot conclude that the 10% uncertainty found by Kylling et al. (2014) would apply here.

ORAC currently allows for the inclusion of two cloud layers in the FM. As there was prevalent stratus cloud during the Raikoke eruption we ran ORAC in both single-layer and multi-layer mode. In multi-layer mode we tried two FM configurations: one with a tightly constrained, low-level (800 hPa) water layer underlying an ash layer and the second with a tightly constrained, mid-level (500 hPa) water cloud underlying an ash layer. We also varied *a priori* settings for the single and multi-layer runs (described in Sect. 3.2.3).

## 3.2 Optimal estimation

The OE technique implemented in ORAC utilises Bayes' theorem so that uncertainties in *a priori* information can be considered in addition to uncertainties (noise) in the satellite measurements (Rodgers, 2000). In practice, the goal of OE is to minimise a cost function that is described by a $\chi^2$ distribution:

$$\chi^2 = [\mathbf{y} - \mathbf{F}(\mathbf{x}, \mathbf{b})]^T \mathbf{S}_\epsilon^{-1} [\mathbf{y} - \mathbf{F}(\mathbf{x}, \mathbf{b})] + [\mathbf{x} - \mathbf{x_a}]^T \mathbf{S_a}^{-1} [\mathbf{x} - \mathbf{x_a}], \tag{6}$$

where $\mathbf{y}$, $\mathbf{x}$ and $\mathbf{x_a}$ are the measurement, state and *a priori* state vectors, respectively, $\mathbf{F}(\mathbf{x}, \mathbf{b})$ is the FM vector (i.e. Eq. 1 converted to brightness temperatures for each satellite channel), which is a function of ancillary information, $\mathbf{b}$, as well as $\mathbf{x}$. Forward model and measurement uncertainties are contained in the measurement error covariance matrix, $\mathbf{S}_\epsilon$, and *a priori* uncertainties are contained in the *a priori* error covariance matrix, $\mathbf{S_a}$. Here it is worth noting that $\mathbf{S}_\epsilon$ and $\mathbf{S_a}$ are assumed diagonal and so all off-diagonal elements (i.e. the covariances) are zero. Therefore, when making this assumption one should be careful to select state variables that are independent of each other. To minimise Eq. 6, the ORAC algorithm uses the well-known Levenberg–Marquardt minimisation scheme (Levenberg, 1944; Marquardt, 1963). Details regarding the implementation of Levenberg–Marquardt for the present study are provided in McGarragh et al. (2018).

### 3.2.1 State and measurement vectors

When constructing state and measurement vectors it is important to consider whether or not the measurements contain enough information about the state that is being retrieved. It is therefore good practice to use at least as many independent measurements as there are state variables. The ORAC state vector contains five state variables: optical depth at 550 nm ($\tau$), effective radius ($r_e$), cloud-top pressure ($p_c$), surface temperature ($T_s$) and cloud fraction ($f$). Experience using ORAC to retrieve $f$ for meteorological clouds has shown that there is a compensating effect if the optical depth and cloud fraction are simultaneously retrieved. Essentially, the optical depth increases as the cloud fraction is reduced (and vice-versa). One approach to addressing this issue is to tightly constrain cloud fraction if it can be estimated from higher resolution data. Watts et al. (1998) and Poulsen et al. (2012) discuss this in more detail. Investigating this issue would be beyond the scope of the present study and so for now we assume a cloud fraction of 1 and account for uncertainty due to cloud inhomogeneity (i.e. broken cloud or pixels at cloud edges), in addition to errors due to the plane-parallel cloud assumption, as a forward model error (see Sect. 3.2.2), which stems from the work of Watts et al. (1998). We therefore do not attempt to retrieve $f$ and assume it is always equal to one. Thus the

state vector used for the ash retrievals presented here contains four state variables:

$$\mathbf{x} = \begin{pmatrix} \log_{10}(\tau) \\ r_e \\ p_c \\ T_s \end{pmatrix}.$$ (7)

Early studies on volcanic ash clouds (e.g. Prata, 1989a; Wen and Rose, 1994; Prata and Grant, 2001) have shown that for a given $T_s$ and $T_c$, $r_e$ varies with the 11-12 μm BTD and $\tau$ varies with brightness temperature measured at 11 μm (or 12 μm). For opaque ash clouds, the BTD will be $\sim$0 K and so there will be no information on particle size in this case. Two-channel thermal infrared retrievals work best when the ash cloud is semi-transparent and there is a strong thermal contrast between the surface and the cloud (Prata and Prata, 2012). The dependence of $r_e$ on the BTD has also been shown for semi-transparent ice and water clouds (Inoue, 1985; Yamanouchi et al., 1987; Prabhakara et al., 1988; Parol et al., 1991; Key, 1995; Cooper et al., 2006; Wang et al., 2011). This explains why OE schemes attempting to retrieve $r_e$ and $\tau$ from thermal-only measurements often include window channels centred near 11 and 12 μm in the measurement vector. Further, as discussed in Pavolonis et al. (2013) and Francis et al. (2012), the addition of the 13.3 μm to the measurement vector improves cloud-top pressure (height) estimation, particularly for optically thin, upper-troposphere/lower-stratosphere (UTLS) ash clouds and the same applies to cirrus clouds (Heidinger et al., 2010). Channels centred near 10.4 μm aboard geostationary imagers (e.g. ABI and AHI) have only recently become available and it is thought that with the inclusion of the 10.4 μm channel to the measurement vector, additional microphysical information on volcanic ash particles may be extracted (Pavolonis and Sieglaff, 2012; Pavolonis et al., 2020). We therefore include AHI channels 10.4, 11.2, 12.4 and 13.3 μm in our measurement vector so that

$$\mathbf{y} = \begin{pmatrix} T_{10} \\ T_{11} \\ T_{12} \\ T_{13} \end{pmatrix},$$ (8)

where $T_{10}$, $T_{11}$, $T_{12}$, $T_{13}$ represent the brightness temperatures in the 10.4, 11.2, 12.4 and 13.3 μm AHI channels, respectively. These channels are advantageous not just for the reasons mentioned above but also because they are unaffected by sulfur dioxide ($SO_2$) absorption, which is not currently included in our FM and was present in abundance for the Raikoke case (Hyman and Pavolonis, 2020; Prata et al., 2021; de Leeuw et al., 2021).

### 3.2.2 Forward model and measurement uncertainty

Forward model uncertainties arise due to assumptions and approximations used to evaluate the FM. Based on previous studies with earlier versions of the ORAC algorithm (Watts et al., 1998, 2011; Poulsen et al., 2012), we assume fixed uncertainty in the thermal channels of 0.50 K which accounts for uncertainties due to the plane-parallel assumption (i.e. 3D radiative effects) and sub-pixel scale inhomogeneity (Iwabuchi and Hayasaka, 2002). We also account for errors due to misalignment (co-registration errors) between channels assuming a fixed uncertainty of 0.15 K for each channel.

To estimate measurement uncertainties in thermal infrared channels, we use the noise-equivalent delta temperature (NEΔT). As mentioned earlier, the NEΔT for satellite imager channels is reported at a particular reference temperature ($T_0$) and thus it is only accurate for that reference temperature. However, it is straightforward to allow error in the measurements to vary with the measured brightness temperature, which recognises that increased signal (i.e. higher brightness temperatures) will result in reduced noise and vice-versa. We therefore compute measurement noise ($\delta T_m$) for each channel based on the measured brightness temperature per pixel as follows

$$\delta T_m = \delta T_0 \left( \frac{\partial B(T_0)}{\partial T_0} \bigg/ \frac{\partial B(T_m)}{\partial T_m} \right), \tag{9}$$

where $\delta T_0$ is the NEΔT reported by the satellite instrument provider and $B(T_0)$ and $B(T_m)$ are Planck functions evaluated at $T_0$ and $T_m$. Figure 1 shows how Eq. 9 allows the measurement noise to vary for a range of brightness temperatures for each channel used in our measurement vector. For reference, the NEΔT for each channel is plotted as a horizontal dashed line to demonstrate how uncertainties using a fixed value lead to underestimations of uncertainty for colder brightness temperatures.

### 3.2.3 Lowest cost, a priori settings and first guess

One of the advantages of the ORAC algorithm is that it allows the user to easily modify cloud layer properties for both single layer and multi-layer FM configurations. Given that there were numerous meteorological clouds (at low- to mid-tropospheric levels) underlying the Raikoke ash clouds, we considered both single-layer ash and multi-layer (ash over water cloud) scenarios. A further consideration was how to deal with local vs. global minima in the cost surface. Considering that the Raikoke ash dispersed into the troposphere and stratosphere (Muser et al., 2020; Horváth et al., 2021b) and that we use thermal-only channels in our retrieval, the height retrievals are strongly-dependent on the temperature profile. Therefore, retrieving heights in the troposphere and stratosphere pose the potential problem of multiple solutions (or multiple minima in the cost surface) due to the inversion of temperature at the tropopause. An additional complication is in the case of an isothermal region in the atmosphere (or a flat cost surface) which is fairly typical of the lower-stratosphere at high latitudes. A nearby radiosonde sounding at around the time of the Raikoke eruption illustrates the problem (Fig. 2).

To address issues related to the retrieval of cloud-top pressure, we ran ORAC using five different configurations representing different choices of the *a priori* pressures in the single and multi-layer FMs. The *a priori* pressure settings were chosen based on CALIPSO observations at the beginning of the eruption (see Prata et al., 2021, Fig. B3(a)) and are summarised in Table 1. For retrieval configurations with a tropospheric *a priori* ash layer pressure, the first guess was set to the pressure level where the measured $T_{11}$ brightness temperature was closest to the ERA5 temperature (searching from the surface to the top-of-atmosphere). For ash layers with an *a priori* pressure level in the stratosphere, the first guess was set equal to the *a priori*. The *a priori* settings for $\tau$, $r_e$ and $T_s$ are summarised in Table 2. The ash layer *a priori* for $\tau$ (at 550 nm) was set to 0.5, which is a typical value for ash cloud retrievals reported in the literature (e.g. Corradini et al., 2016). For the ash layer *a priori* effective radius, we set $r_e$ to 5 μm, which corresponds roughly to the centre of particle sensitivity for thermal IR channels (Prata and Grant, 2001). The ash layer *a priori* uncertainties on $\tau$ and $r_e$ were set to a large number ($1 \times 10^8$) to ensure that these parameters were effectively unconstrained by their *a priori* values. The *a priori* values for $\tau$ and $r_e$ for water cloud were

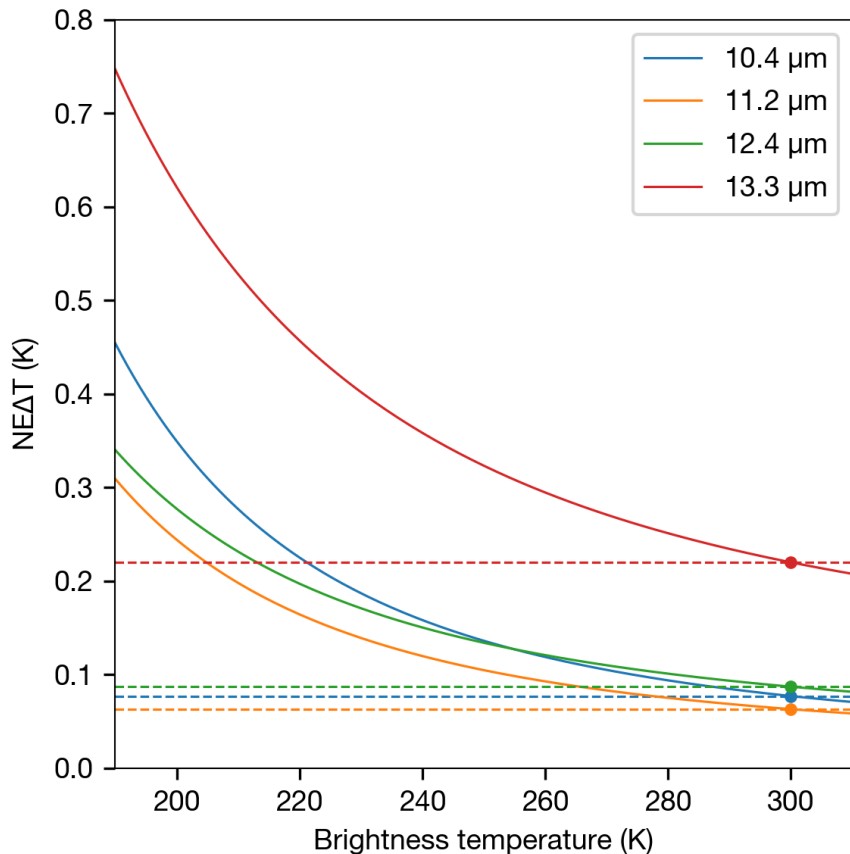

**Figure 1.** Relationship between fixed noise uncertainty values (dashed lines) and varying uncertainty noise values (solid lines) derived from Eq. 9. Solid circles indicate reference temperatures ($T_0$) given for the NE$\Delta$T estimate.

chosen based on ORAC cloud retrievals (Poulsen et al., 2012; McGarragh et al., 2018) applied to a stratus deck close to the ash cloud at the beginning of the Raikoke eruption. The water layer *a priori* uncertainties for $p_c$, $\tau$ and $r_e$ were tightly constrained such that the measurements only influenced the solution for the upper ash layer in the multi-layer runs.

After all five retrieval configurations were run, the retrieval configuration which resulted in the lowest cost was selected on a per-pixel basis to generate the final retrieval product. Figures 3(a)–(e) show the cost at measurement solution at 23:00 UTC on 22 June 2019 for each retrieval configuration on a per-pixel basis and the resulting cost map when the minimum cost from each configuration is selected (Fig. 3(f)). For this scene, the differences in the five cost values vary over several orders of magnitude in some parts of the ash cloud but can be quite similar (same order of magnitude) in other parts. The most notable differences in cost amongst the five retrieval setups are seen when comparing the stratospheric *a priori* retrieval configurations to the tropospheric *a priori* retrieval configurations. To illustrate the relative differences in the five cost values, we have generated a 'forward model flag' where each pixel is coloured according to the retrieval configuration that resulted in the lowest cost. The

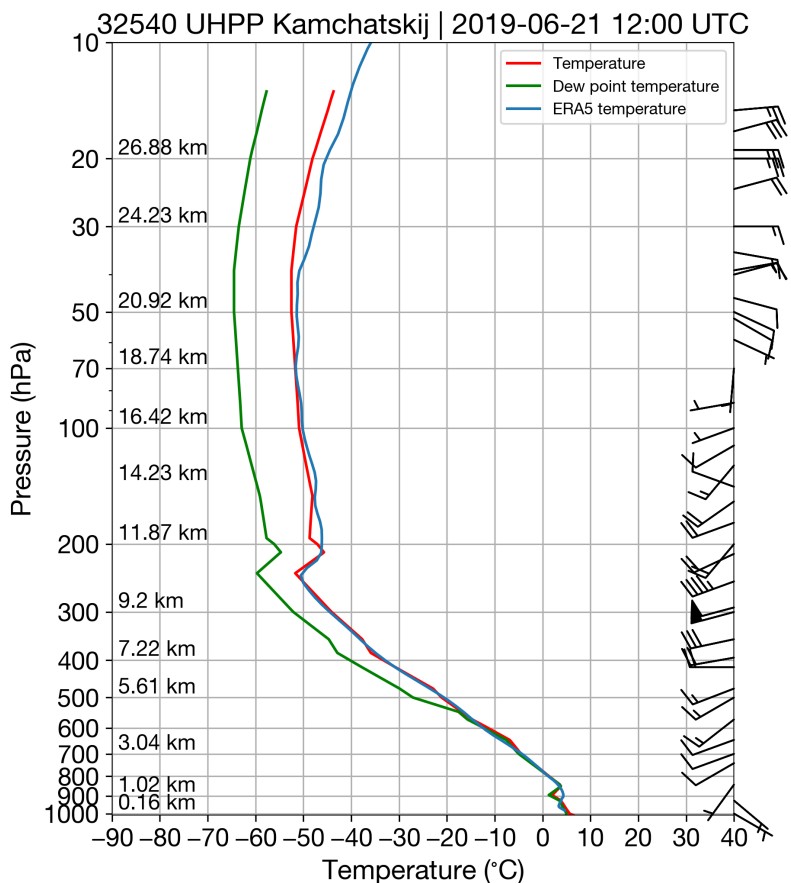

**Figure 2.** Radiosonde sounding from Kamchatskij station (53.08° N, 158.58° E, 84 m) at 12:00 UTC on 21 June 2019. Data accessed from University of Wyoming sounding database (last access 26 February 2022). ERA5 temperature for the grid-box corresponding to Kamchatskij station's location and sounding time is over-plotted in blue.

$T_{11}$ brightness temperature (Fig. 3(g)) and the natural-colour composite (Fig. 3(h)) provide contextual information and show that the selected forward model configuration's are reasonable. For example, the stratospheric ash over 500 hPa water cloud configuration (light grey pixels) returned the lowest cost for a part of the ash cloud overlying the cold (high) cloud associated with the cyclone.

In summary, this approach accounts for multi-layer cloud scenarios, multiple local minima in the cost function (in the troposphere and stratosphere) and reduces the impact of flat cost surfaces (isothermal regions) with the use of the *a priori* uncertainty settings.

**Table 1.** *A priori* and first guess settings for $p_c$ for both multi-layer and single layer forward model configurations. For the mult-layer configurations, the first number in the *a priori* column corresponds to the pressure level of ash and the second to the water cloud. Lower and upper limits within ORAC for $p_c$ are 10 hPa and 1200 hPa, respectively.

| Forward model | *A priori* | First guess | Uncertainty |
|---|---|---|---|
| Ash single layer | 500 hPa | ERA5 | 200 hPa |
| Ash single layer | 200 hPa | 200 hPa | 200 hPa |
| Ash above water | 500 hPa / 800 hPa | ERA5 / 800 hPa | 200 hPa / 50 hPa |
| Ash above water | 200 hPa / 800 hPa | 200 hPa / 800 hPa | 100 hPa / 50 hPa |
| Ash above water | 200 hPa / 500 hPa | 200 hPa / 500 hPa | 100 hPa / 50 hPa |

**Table 2.** *A priori* settings for $\tau$, $r_e$ and $T_s$ for the ash and water layers considered. All first guesses for these parameters were set to be equal to their *a priori* values.

| Parameter | $\tau_{\text{ash}}$ | $r_{e,\text{ash}}$ (μm) | $\tau_{\text{wat}}$ | $r_{e,\text{wat}}$ (μm) | $T_s$ (K) (sea/land) |
|---|---|---|---|---|---|
| *A priori* | 0.5 | 5 | 16 | 10 | ERA5 |
| Uncertainty | $1 \times 10^8$ | $1 \times 10^8$ | 2 | 1 | 2.0/5.0 |
| Lower limit | 0.001 | 0.01 | 0.001 | 0.1 | 200 |
| Upper limit | 255.9 | 20 | 255.9 | 35 | 400 |

## 3.3 Ash mass loading and uncertainty

To compute ash mass loading (mass per unit area) we use the standard formulation used by many previous authors (Wen and Rose, 1994; Prata and Grant, 2001; Corradini et al., 2008; Pavolonis et al., 2013):

$$m_l = \frac{4}{3} \times \frac{\tau \cdot r_e \cdot \rho}{Q_{\text{ext}}}, \tag{10}$$

where $\tau$ is the optical depth at 550 nm, $Q_{\text{ext}}$ is the extinction efficiency factor for ash at 550 nm ($Q_{\text{ext}} \approx 2$ at 550 nm) and $\rho$ is the ash particle density, assumed to be 2300 kg m$^{-1}$. We assumed an ash particle density of 2300 kg m$^{-1}$ to be consistent

with what is used in the Numerical Atmospheric-dispersion Modelling Environment (NAME) model as an earlier version of the retrievals presented here were used for comparison to the model in Harvey et al. (2022). This ash density was determined for operational use and is therefore a representative average value (Witham et al., 2019). We have accounted for uncertainty in this value by allowing for an absolute uncertainty of 300 kg m$^{-1}$. It is interesting to note that the definition of ash mass loading

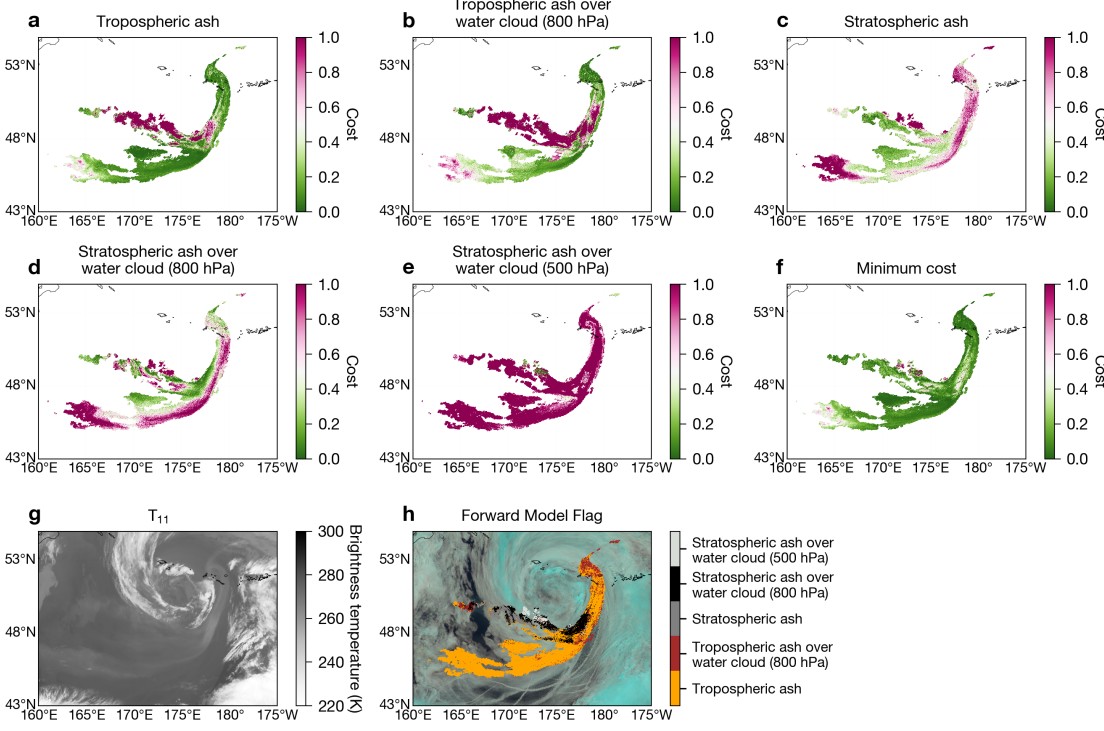

**Figure 3.** (a)–(e) Measurement cost at solution for each of the forward model configurations (annotated above each subplot) used in the present study. (f) Minimum cost per pixel out of the five configurations shown in (a)–(e). (g) 11.2 μm brightness temperature. (h) Forward model flag (i.e. forward model configuration that resulted in the lowest cost per pixel). Natural colour composite is plotted beneath for context.

(Eq. 10) is analogous to the cloud liquid-water path found in numerous cloud retrieval studies (see Eq. 24 of Poulsen et al., 2012, for example). To compute uncertainty on the ash mass loading, $\Delta m_l$, we assume all variables are independent and sum the error terms in quadrature:

$$\left(\frac{\Delta m_l}{m_l}\right)^2 = \left(\frac{\Delta \tau}{\tau}\right)^2 + \left(\frac{\Delta r_e}{r_e}\right)^2 + \left(\frac{\Delta \rho}{\rho}\right)^2 + \left(\frac{\Delta Q_{\text{ext}}}{Q_{\text{ext}}}\right)^2. \tag{11}$$

Note here all error terms are retrieved, except for the ash particle density ($\Delta \rho$) and $\Delta Q_{\text{ext}}$, which is assumed to be negligible compared to the other error terms.

## 3.4 Ash detection flag

By default ORAC is run on every level 1b satellite pixel in the full disk image. In some circumstances retrievals can converge, albeit with a poor fit to the measurements (high cost), even when the FM is not representative of the observation. To avoid these situations and speed up processing times, we only considered pixels within a spatial region from 135° E–15° W and 40–65° N

that had an 11-12 μm BTD of less than 0.5 K, which is a fairly loose constraint for ash detection. Pixels were then flagged as
'ash' if the water vapour corrected BTD, $\Delta T_{ash}$, was less than -0.20 K. The water vapour correction was applied following the
approach of Yu et al. (2002). After this initial ash detection threshold was applied there were generally two cases that resulted
in false positives: (1) surface inversions and (2) inversions above cloud-tops. We removed these cases by setting the pixel to
'ash free' if:

$$-1.25 < \Delta T_{ash} < -0.20 \text{ and } T_{11} > 275 \text{ K} \tag{12}$$

or

$$-0.40 < \Delta T_{ash} < -0.20 \text{ and } T_{11} < 240 \text{ K}. \tag{13}$$

These thresholds were chosen based on manual inspection of the data. A further step to improve the 'ash flag' was to apply
an 'opening' morphological 3x3 spatial filter designed to remove isolated pixels unrelated to the ash cloud/plume. Finally all
pixels with satellite zenith angles greater than 75 ° were ignored as the plane-parallel assumption breaks down at extreme
satellite view angles.

### 3.5  Quality control

To ensure that only the highest quality retrievals were considered for scientific interpretation, we ran a quality control test on
each pixel identified by the ash flag (Sect 3.4). The quality control checks that the retrieval converged, all state variable relative
uncertainties were not greater than 100 % and that the retrievals were within a physically sensible range. For the present
analysis, the range of valid values considered were 0–15 μm, 0–20 and 0–35 km for $r_e$, $\tau$ and $h_c$ (cloud-top height converted
from $p_c$), respectively.

### 3.6  Gap filling

After quality control we noticed that 'gaps' in the retrieval fields were appearing where the ash flagging had originally detected
ash and that the number of gaps varied with time. In general, the number of gaps increased as the total number of ash-
contaminated pixels increased. Given that we have good information (quality-controlled retrievals) adjacent to these gaps, we
implemented an algorithm that aims to fill the gaps in the retrieved fields. Specifically, we implemented the Qhull algorithm
(Barber et al., 1996), which identifies the convex hull of a set of arbitrary points. After finding the convex hull, Delaunay
triangulation is used to perform linear barycentric interpolation to fill the missing data. At certain times, the fraction of gap
filled pixels can be significant (reaching as high as ~34 %). Figures 4(a) and (b) show the ash mass loading at 08:00 UTC on
23 June 2019 before and after gap filling (at this time the percentage of gap filled pixels is 23%). It is important to note that at
the time when the total mass of very fine ash reached its maximum, the fraction of gap filled pixels was ~7 %, meaning that,
regardless of gap filling, the maximum total mass estimate is within the uncertainty range estimated here (see Sect. 4.1). The
gap filling algorithm was applied to all retrieved state variables and associated uncertainty fields as well as the mass loading
and uncertainty computed from Eqs. 10 and 11, respectively.

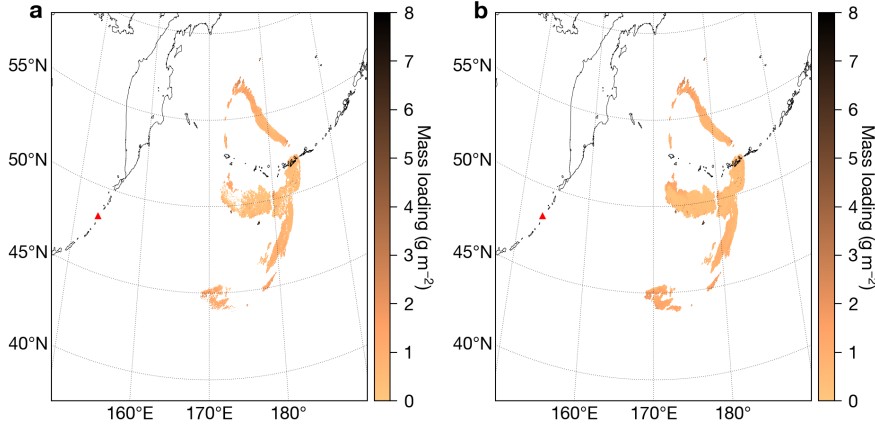

**Figure 4.** (a) Mass loading retrieval before gap filling processing step. (b) Mass loading after gap filling processing step.

## 3.7 Parallax correction

Due to the high satellite zenith angles (∼55–75°) and high altitude of the volcanic clouds (>10 km) we also needed to correct the retrievals for parallax. We followed the method of Vicente et al. (2002) to compute the latitude/longitude parallax shift based on the ORAC retrieved height. We applied the parallax correction to all of the gap filled, quality-controlled retrievals. The parallax correction resulted in pixel shifts as high as ∼ 20 km for observations at the beginning of the eruption (e.g. Fig. 5(a)). For cases where the parallax shift resulted in two solutions in the same pixel, the set of retrieved fields which corresponded to the higher height was selected. In some instances the parallax shift can leave behind gaps within the boundaries of the volcanic cloud meaning that these pixels are being obscured by other parts of the plume. Note that the magnitude of the shift is dependent upon the retrieved height (the shift increases with increasing height for the same viewing angle). To address this, we filled the parallax correction gaps by first applying a 2x2 'closing' morphological filter to the parallax-corrected ash flag and then filled the retrieval fields using the Qhull algorithm as before (Sect. 3.6).

## 4 Results and discussion

### 4.1 Raikoke source term and long-range ash transport

The June 2019 eruption from Raikoke volcano (48.292° N, 153.25° E, 551 m) is described in detail by McKee et al. (2021). Here we provide an overview of the volcanic ash emissions based on AHI satellite measurements and the ORAC retrieval results. According to AHI measurements the Raikoke eruption began at around 18:00 UTC on 21 June 2019. What followed was a series of explosive eruptions characterised by sharp decreases in brightness temperatures over the volcano. Nine explosive eruptions can be clearly identified in the AHI time-series data with several smaller events more easily identified with the aid of true colour and thermal imagery.

Figure 5 shows a high resolution (10-minute) time-series of the ORAC cloud-top height retrievals for the Raikoke eruption sequence where we have relaxed the BTD thresholds (i.e. removed the threshold conditions in Eqs. 12 and 13) to retrieve the height in the opaque parts of the plume (Fig. 5(a)). Note that the effective radius retrievals within the opaque regions of the cloud are not reliable when using thermal-only measurements as BTDs close to zero mean that the solution space cannot be interpolated (i.e. there is no information on particle size for opaque plumes). The time-series shows maximum heights (and associated uncertainty) within a search radius of 7.5 km from the volcano (Fig. 5(b)). The parallax shift is large (∼20 km) for the initial Raikoke plume (grey-shaded regions), demonstrating that without a parallax correction significant errors could be introduced when comparing height and location (latitude/longitude) to other satellite datasets or dispersion model output. To compare the ORAC cloud-top height retrievals to the GOES-17 side-view heights, we varied the search radius until the optimal 7.5 km radius was found. We define 'optimal' as the search radius that resulted in the closest match (minimised sum of squared differences) to the GOES-17 side-view height data. This approach ensures a robust comparison to the GOES-17 data as the exact coordinates of the side-view heights are not provided in the Supp. Matt. of Horváth et al. (2021b). The first six pulses were short-lived (eruption duration from 10-40 minutes) with the seventh being the largest continuous ash emission, lasting almost 4 h. Following a pause of ∼1 h, two large explosive eruptions occurred at around 03:40 UTC and 05:30 UTC on 22 June 2019. The maximum height retrieved during the explosive phase of the eruption was 14.2 km (asl) with numerous pulses injecting ash into the stratosphere (tropopause was typically 11.2 km; Fig. 5(b)). The major phase of the eruption had subsided by 10:00 UTC on 22 June. Based on the ORAC time-series we estimate an eruption duration of 11.7 h and a median plume-top height of $10.7 \pm 1.2$ km (asl).

The GOES-17 side-view retrievals, with an estimated uncertainty of $\pm$ 500 m (Horváth et al., 2021a), serve as validation for heights estimated with ORAC for the initial eruption. The ORAC heights mostly agree well with the GOES-17 side-view heights, with some notable disagreement during the largest duration eruption and the final two pulses (Fig.5(b)). These differences are not surprising as the heights reported at these times using the GOES-17 side-view method correspond to small-scale turrets overshooting the main umbrella (see Supp. Mat. of Horváth et al., 2021b, for details). While ORAC does allow for overshoots by extrapolating the lapse rate (determined from two levels just beneath the tropopause) into the stratosphere (McGarragh et al., 2018), the retrieval is relying on thermal infrared measurements that are coarser in spatial resolution compared to the visible channels used by the GOES-17 side-view height analysis. The ORAC heights are, however, an improvement to simply matching a brightness temperature to a NWP profile. Figure 5(b) shows the cloud-top heights derived using the minimum brightness temperature method described in McKee et al. (2021) (hereafter '$BT_{min}$ method'). Note that McKee et al. (2021) used ERA-Interim data whereas we use ERA5 here. The ORAC and $BT_{min}$ methods give quite similar results when compared to the GOES-17 heights (similar correlation coefficient and precision). However, there is a significant bias of -1.92 km for the $BT_{min}$ method, while the ORAC CTH bias compared to GOES-17 is -0.75 km. This result is to be expected because the $BT_{min}$ method will, by definition, never produce heights in the stratosphere. While ORAC effectively matches the height to the temperature profile, it accounts for other factors (e.g. cloud emissivity/transmission, viewing angle, above cloud transmission, information from the 13.3 μm channel) compared to matching the raw 11 μm brightness temperature. Further, ambiguity as-

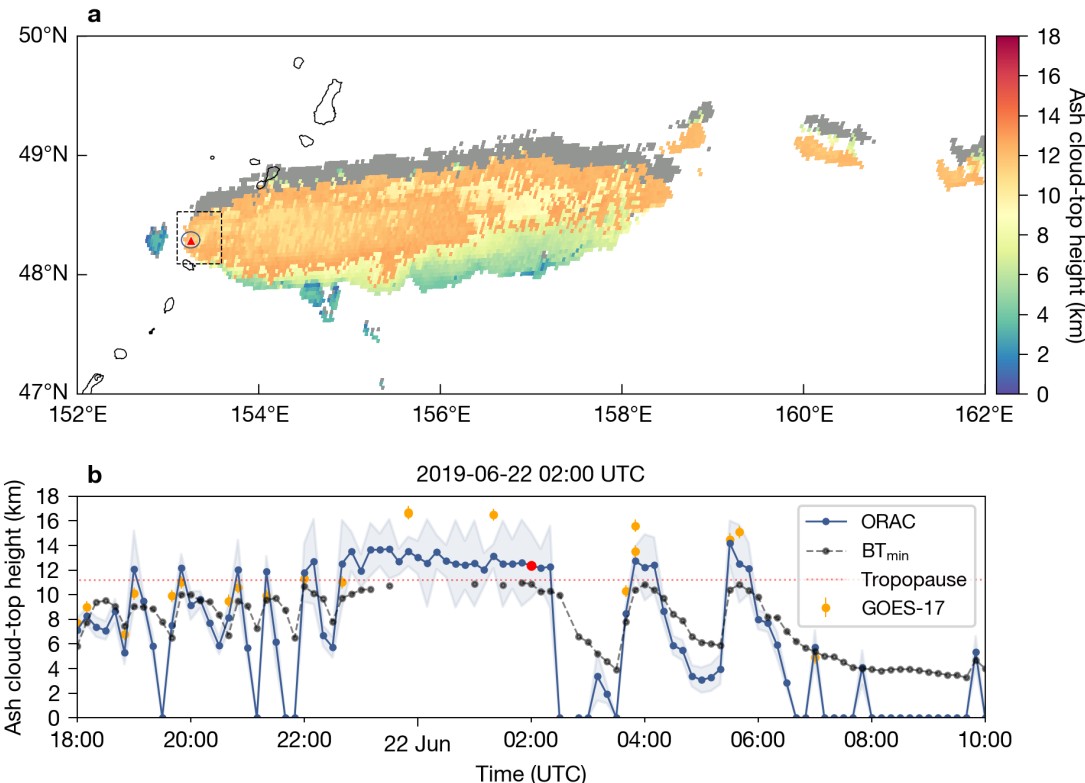

**Figure 5.** (a) Parallax-corrected ORAC ash cloud-top height retrievals at 02:00 UTC on 22 June 2019. Grey shaded regions indicate parallax shift. Blue circle indicates a 7.5 km radius around the volcano (red triangle). Black dashed box indicates search region used to find minimum brightness temperatures in McKee et al. (2021). (b) High temporal resolution (10-minute) time-series of ORAC ash cloud-top height retrievals for the Raikoke eruption. Ash cloud-top height for the ORAC retrievals (blue filled circles) in the time-series corresponds to the maximum height within at 7.5 km radius of the volcano. Uncertainties associated with the ORAC heights are indicated as light blue shading around the blue filled circles. Black filled circles indicate cloud-top heights determined by matching the minimum brightness temperature within the black dashed bounding box in (a) to the closest ERA5 grid-box profile to the volcano. ERA5 temperature profiles were linearly interpolated vertically and in time to match the minimum AHI brightness temperature every 10 minutes (i.e. 'BT$_{min}$' method). Orange filled circles indicate GOES-17 side-view heights taken from Horváth et al. (2021b). Orange error bars indicate $\pm$ 500 m. Red dotted line indicates lapse rate tropopause.

sociated with multiple height solutions (due to temperature inversions) is accounted for using optimal estimation by selecting the lowest cost from the output of the five ORAC model run configurations (described in Sect. 3.2.3).

Figure 6 shows four scenes (observation times) that illustrate the time evolution and long-range transport of the Raikoke ash cloud. The ORAC mass loading (Fig. 6(a)) and cloud-top height (Fig. 6(b)) retrievals show that ash dispersed primarily to the east with a tropospheric branch of the ash cloud separating toward the south and a stratospheric portion of the ash

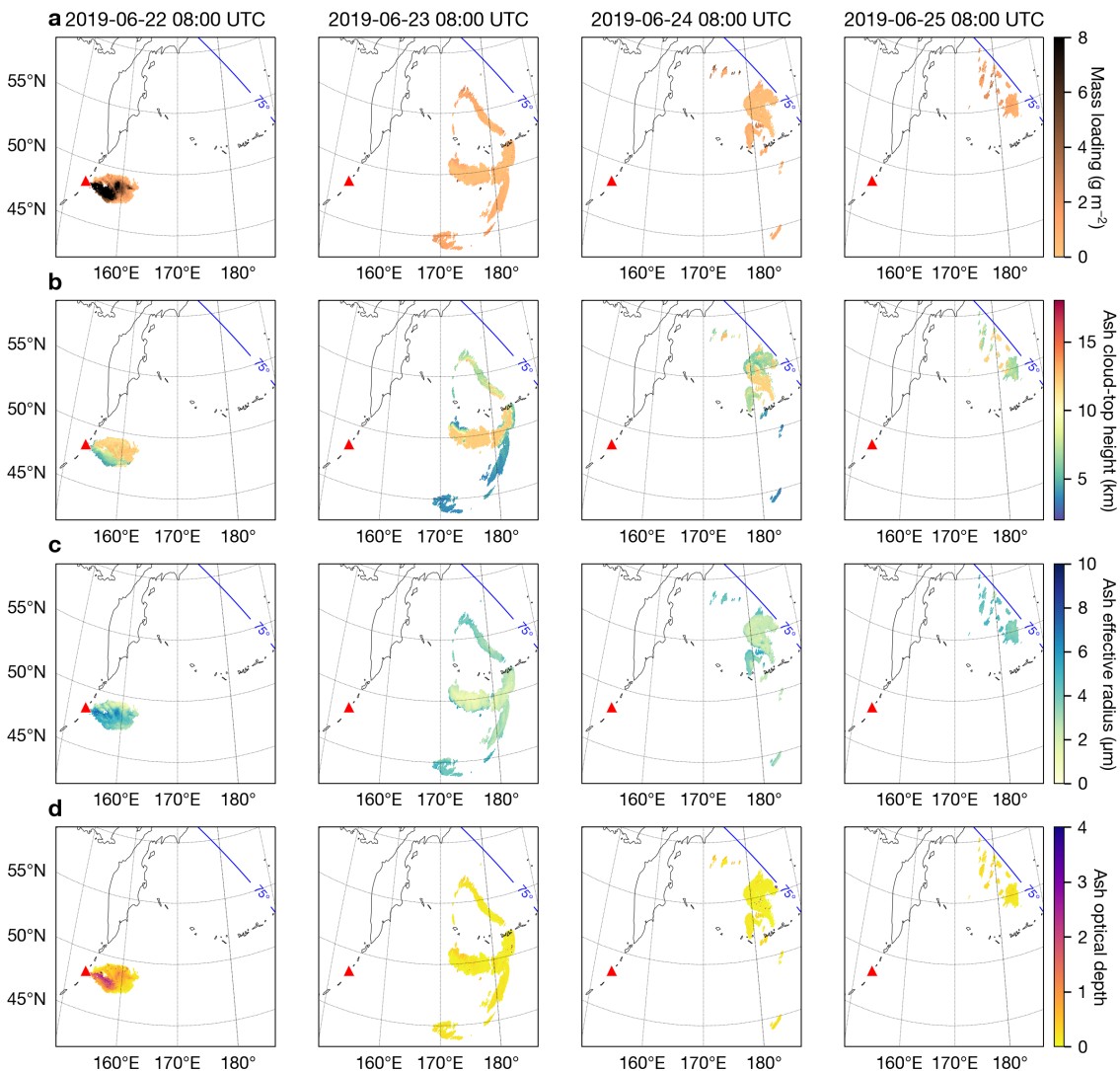

**Figure 6.** ORAC retrievals for (a) mass loading, (b) ash cloud-top height, (c) ash effective radius and (d) ash optical depth at 550 nm. Observations times annotated at the top of each column in the plot. Blue line on each plot indicates the AHI satellite zenith angle of 75 °.

cloud wrapping up in a cyclone eventually resulting in ash transport toward the north. The ORAC retrievals show that the tropospheric branch of the ash cloud generally had higher mass loadings (10–30 $\mathrm{g\,m^{-2}}$) and effective radii (4–7 µm) whereas the stratospheric portion had comparatively lower mass loadings (1–10 $\mathrm{g\,m^{-2}}$) and effective radii (0.5–2.5 µm). The tropospheric ash cloud maintained heights of 5–8 km whereas stratospheric ash cloud-top heights remained just above the tropopause at 11–13 km. At 08:00 UTC on 23 June (Fig. 6(b) second column) there are three distinct levels in the ash cloud; a mid-tropospheric portion (3.5–4 km), a longitudinally-extended, lower-stratospheric region (12 km) and an upper-tropospheric region (6–8 km).

By 08:00 UTC on 24 June the mid-tropospheric portion of the ash cloud is no longer detectable with the ash flag, while
the upper-tropospheric and lower-stratospheric regions continued to disperse toward the north reaching extreme satellite view
angles (where the ORAC retrievals are not possible) by 08:00 UTC on 25 June.

Figure 7(a) shows the time-series of total mass of very fine ash from 21 June at 18:00 UTC to 25 June at 18:00 UTC.
Although our analysis period covers seven days, we found that the retrievals began to fail or gave spurious results after four
days, suggesting that median 550 nm optical depths of $\sim$0.1 (Fig. 7(b)) are at or close to the detection limit for the ORAC-ash
retrievals applied to AHI. The peak of the total mass time series was reached $\sim$13 h after the eruption began (at 07:00 UTC on
22 June 2019) with a total mass of $0.73 \pm 0.40$ Tg. After reaching its maximum, the total mass decreases with an $e$-folding time
of $\sim$20 h. From 24 June onward, approximately 0.1 Tg of ash remained in the atmosphere based on the present ash detection
and retrieval scheme. The ORAC estimate of the total mass of very fine ash is somewhat lower than existing estimates that also
use AHI data to retrieve the total mass (e.g. $1.1 \pm 0.7$ Tg in Muser et al., 2020). One reason for this is due to the difference
in ash particle density assumed here ($2300 \pm 300$ kgm$^{-3}$) and what was assumed in Muser et al. (2020) (2600 kgm$^{-3}$). Other
differences include differing refractive index data, size distribution assumptions, ash detection thresholds, the assumption of a
single layer of ash vs. multi-layer cloud/ash scenarios and the use of two channels to retrieve optical depth and effective radius
compared to the four channels used in the present study.

The time-series of median effective radius is shown in Fig. 7(c) and reveals that larger particles ($\sim$6 μm) dominated the
plume during the first $\sim$12 hours with a transition to smaller particles ($\sim$3 μm) from 12–24 h after eruption. After 60 h post-
eruption there is an apparent increase in median effective radius from 3–4 μm. However, given the median satellite zenith
angles (Fig. 7(c) right axis) were $>70°$ at this time, it is likely that this increase is due to a retrieval artefact rather than a real
increase in particle size in the volcanic cloud. Gu et al. (2005) also found that geostationary satellite retrievals at zenith angles
of $\sim$70 degrees resulted in an overestimation of effective radius when compared with close-to-nadir (MODIS and AVHRR)
retrievals. Factors contributing to retrieval artefacts include: violation of the plane parallel cloud assumption, 3D radiative
transfer effects, limb darkening and pixel distortion due to extreme viewing geometry. Parts of the ash cloud may also be
exiting the AHI field-of-view at this time, which could also lead to an abrupt change in the median effective radius.

The time-series for median cloud-top heights in the troposphere and stratosphere are shown in Fig. 7(d). We chose to plot the
median cloud-top heights separately for the troposphere and stratosphere due to the distinct levels of ash that formed following
the eruption. Cloud-top heights in the troposphere generally increased from 5–8 km in the first 12 h, followed by a decrease to
5 km after 24 h of atmospheric residence. After this period much of the ash had fallen out or was undetectable (cf. Fig. 7(a))
with the median tropospheric heights varying from 5–7 km (asl). The median stratospheric cloud-top heights remained fairly
constant at 12 km throughout our analysis period with some minor variation (11-13 km). These heights are close to the chosen
*a priori* stratospheric height (200 hPa) and probably reflect the fact that the measurements had little influence on the retrieved
solution due to the isothermal nature of the lower-stratosphere (Fig.2).

## 4.2 Total mass erupted and distal fine ash mass fraction

The total mass of very fine ash (radius $\leq 15$ µm) is an important piece of information for dispersion modellers attempting to forecast long-range, fine ash transport. The London VAAC parameterises the source term using an estimate of the plume-top height above vent level ($H$) converted to a mass eruption rate ($\dot{M}$) using the empirical fit determined by Mastin et al. (2009). The relationship has seen wide usage in the scientific literature due to its simplicity and the fact that it only requires one input readily determined from satellite data. This power law relationship relates the total mass erupted (all particle sizes) to the maximum plume height (above vent level) and therefore the distal fine ash mass fraction needs to be set in order to simulate very fine ash transport and dispersion. Typical distal fine ash mass fractions used by the London VAAC range from 1-5% (Dacre et al., 2013). Based on a comparison between well-constrained values of $\dot{M}$, eruption duration and satellite retrievals, Gouhier et al. (2019) found that the distal fine ash mass fraction varies by $\sim$2 orders of magnitude (0.1–6.9%) and decreases with increasing $\dot{M}$. At the wavelengths used here the retrievals effectively measure the volume of the volcanic ash. For the distal plume, this signal is dominated by fine ash effective radii in the range 0.5–9 µm. For the Raikoke eruption, Osborne et al. (2022) report a total mass of 300 Tg based on the Mastin relationship ($\dot{M} = 140.8H^{1/0.241}$), assuming a plume-top height of 15 km and continuous emission from 21:00 UTC on 21 June to 03:00 UTC on 22 June (eruption duration of 6 h). It is not clear how they arrived at the 300 Tg figure, as a constant plume height of 15 km (asl; 14.45 km above vent level) for a 6 h duration equates to a total mass of 198 Tg when using Mastin et al.'s empirical fit. Osborne et al. (2022) caution that their total mass estimate should only be taken as a 'representative figure' that is likely an upper-bound on the total mass due to the complexity of the eruption source (i.e. numerous pulses, pauses and variations in plume height). The GOES-17 side-view times-series data show that the plume heights varied with time from 9–14 km with overshoots reaching 15–16 km. It is likely that using a constant height of 15 km (asl) will result in a significant overestimate of the total mass, especially considering the power law relationship between $\dot{M}$ and $H$. Despite this, the Osborne et al. (2022) estimate is closer to the lower range of McKee et al. (2021) who determined total masses ranging from 287–672 Tg (average value of 439 Tg) based on more sophisticated plume modelling constrained with plume heights ranging from 10–12 km (asl). A possible reason for this discrepancy may be due to the fact that Osborne et al. (2022) have underestimated the eruption duration. They assume that the Raikoke eruption started at 21:00 UTC on 21 June and ended at 03:00 UTC on 22 June, which neglects five significant eruptions at around 18:00, 18:50 and 19:40 UTC on 21 June and at around 03:40 and 05:30 UTC on 22 June (see Fig. 5(b)). McKee et al. (2021) derive a more detailed eruption sequence of events, showing eruptive activity (recorded by infrasound, lightning and AHI data) from around 18:00 UTC 21 June to 10:00 UTC on 22 June. However, those authors note that it is possible they have systematically underestimated the plume-top height for the Raikoke eruption. They select the coldest pixel from $T_{11}$ and match it to the temperature corresponding to the highest height below the tropopause (using ERA-Interim data), despite multiple lines of satellite-based evidence showing ash in the stratosphere at the beginning of the eruption. Therefore, the total mass estimates from Osborne et al. (2022) and McKee et al. (2021) broadly agree but this could be because of compensating errors; Osborne et al. (2022) overestimate the plume height and underestimate the source duration whereas McKee et al. (2021) underestimate plume-top height.

To understand how the ORAC retrievals translate and compare to existing estimates of the total mass, we converted the time-series of ORAC heights (Fig. 8(b)) to a time-series of $\dot{M}$ based on the Mastin relationship. We then estimated the total mass by integrating over the $\dot{M}$ time-series. After propagating the ORAC-derived uncertainties in height through the Mastin equation (see Appendix A), we obtain an estimate of $101 \pm 67$ Tg for the total mass erupted. Comparing this figure to the maximum total mass of very fine ash derived from the ORAC mass loading retrievals, we estimate a distal fine ash mass fraction of

$0.73 \pm 0.62$ %. Distal fine ash fractions of this magnitude are lower than what is currently used by the London VAAC (1-5%) and adds support to the Gouhier et al. (2019) finding that distal fine ash mass fractions should be set depending on the eruption style.

## 4.3   Ash mass loadings vs ash concentrations

Volcanic ash mass loadings represent the column mass per unit area and are distinct from ash concentrations (mass per unit

volume). However, while passive imager measurements do not resolve the ash layers vertically, the mass loadings can be converted to ash concentrations if a geometric thickness is assumed or measured (Sears et al., 2013). Prata and Prata (2012) demonstrated that ash concentrations could be derived from ash mass loading retrievals when combined with vertically resolved measurements from CALIOP. While only a few intersections have been identified for CALIOP for the Raikoke eruption, previous studies provide an indication of typical ash cloud geometric thicknesses. Prata et al. (2015) found that ash clouds

produced by the 2008 Chaitén eruption were $\sim$0.3–0.7 km thick according to CALIOP observations. Winker et al. (2012) presented CALIOP observations of the 2012 Eyjafjallajökull ash clouds and derived geometric thicknesses of 0.4–1 km. Prata et al. (2017a) studied both ash and sulfates and demonstrated that geometric thicknesses for the ash clouds produced by the 2011 Puyehue-Cordón Caulle eruption were $1.82 \pm 0.55$ km. Based on comparisons made with CALIPSO in the present study (Sect. 4.4), the Raikoke ash cloud geometric thicknesses were $1.04 \pm 0.56$ km. Figure 8 shows the median ash concentrations

that would be derived if geometric thicknesses of 0.5, 1 and 2 km were assumed. The time-series demonstrates that after 16 h the median ash concentration (for all geometric thicknesses assumed) would fall below what ICAO regards as a 'high' ash concentration level ($4 \mathrm{\,mg\,m^{-3}}$).

## 4.4   Cloud-top height validation

Figure 9(a) and (b) show the validation results for the ORAC ash cloud-top heights compared against GOES-17 and CALIOP.

In total we found 115 collocations between the ORAC heights and the validation data derived from CALIOP and GOES-17. There is generally good agreement between the ORAC heights and the GOES-17 data for heights retrieved in the near-source plume at the beginning of the eruption ($R = 0.84$, bias = -0.75 km). However, there is a notable negative bias (-2.67 km) and poorer agreement ($R = 0.67$) for the ORAC heights retrieved in the distal ash clouds when compared against CALIOP. In both comparisons (Figs 9(a) and (b)), ORAC showed limited skill in retrieving heights in the stratosphere. Negative biases

between thermal IR height retrievals and lidar-derived heights have been observed before (Pavolonis et al., 2013; Francis et al., 2012) and is explained by the fact that the effective thermal emission height of the cloud is generally lower than the cloud-top detectable by lidar backscatter measurements. We tried to account for this by using the CALIOP mid-layer heights (rather

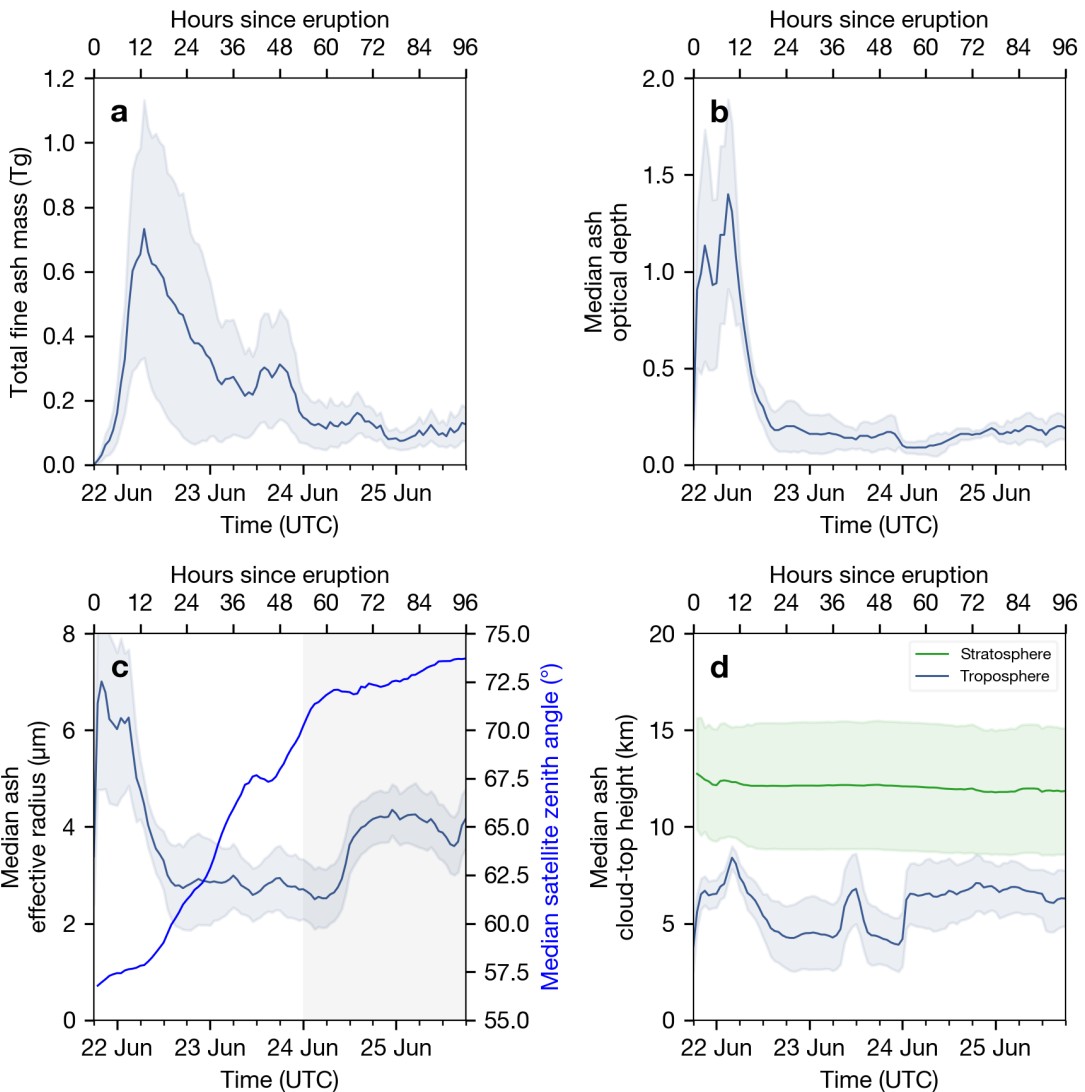

**Figure 7.** Time evolution of (a) total very fine ash mass, (b) median ash optical depth at 550 nm, (c) median ash effective radius and (d) median ash cloud-top height for the first four days of atmospheric residence following the Raikoke eruption. Shaded region in (a) represents the total mass uncertainty computed from the mass loading uncertainty fields (Eq. 11) and shaded regions in (b), (c) and (d) represent the median of the uncertainty in $\tau$, $r_e$ and $h_c$, respectively. Right axis of (c) shows the time-series of the median satellite zenith angle (bright blue line) and shaded region indicates times where median zenith angle exceeds 70 °.

than layer-top heights) from the upper-most layer in the MLay product; however, the negative bias persisted. The precision (standard deviation of the difference between the ORAC and validation heights) for the near-source plume heights (GOES-540 17 comparison) is 1.78 km, which is comparable to but somewhat higher than existing thermal IR-height retrieval schemes

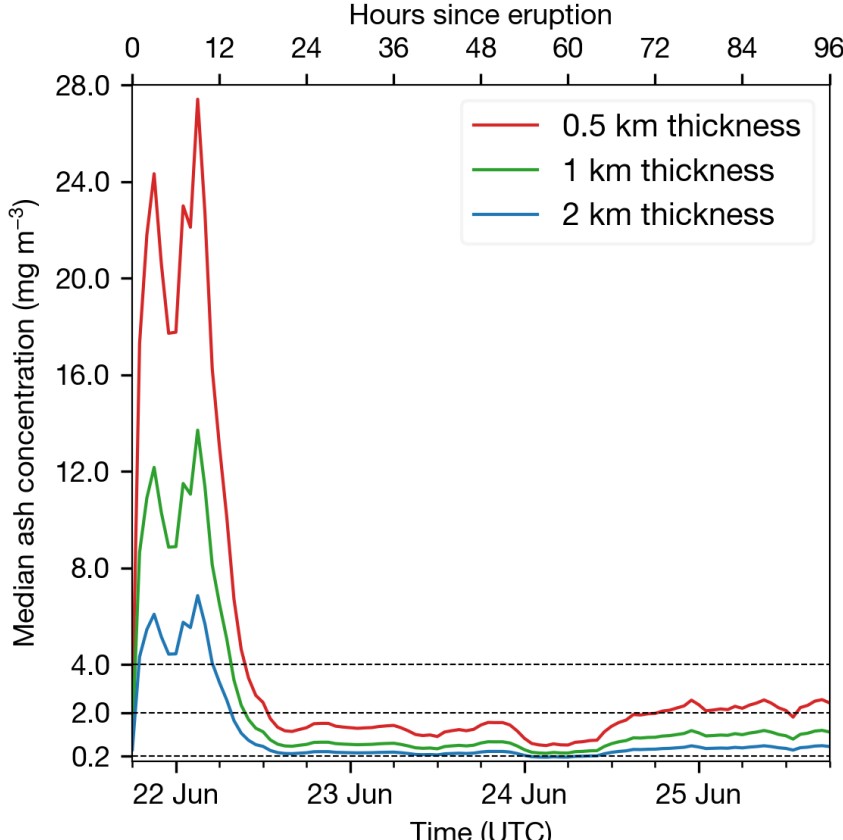

**Figure 8.** Time evolution of median ash concentrations produced by the Raikoke eruption assuming different geometric thicknesses. The ICAO peak ash concentration safety limits at 2 and 4 $mg\,m^{-3}$ are indicated by dashed horizontal lines. The 0.2 $mg\,m^{-3}$ horizonal dashed line indicates the Prata and Prata (2012) detection limit (0.2 $g\,m^{-2}$) converted to an ash concentration assuming a 1 km geometric thickness.

(cf. 1.48–1.64 $km$; Pavolonis et al., 2013). However, a key difference here is that we are validating height retrievals in the stratosphere and troposphere whereas previous validation studies only considered height retrievals in the troposphere. The precision of ORAC heights for the distal ash (CALIOP comparison) was higher (2.75 $km$) than the precision of ORAC heights retrieved for the near-source plume.

Figure 10 shows a CALIPSO overpass where we identified the largest number of height collocations with the ORAC height retrievals. This observation also serves as an important test case for height retrievals in the troposphere and stratosphere. In general, the height retrievals in the troposphere are underestimated (reflecting the negative bias seen in Fig.9(b)). The stratospheric height retrievals show very good agreement with the CALIOP observations; however, some ORAC height retrievals returned heights in the mid-troposphere (5–6 $km$) when there was a stratospheric feature in the CALIOP data from 12–13 $km$.

For these cases, the CALIOP 'Feature Optical Depth' at 532 nm for the top layer was less than 0.05 and so it is likely that the

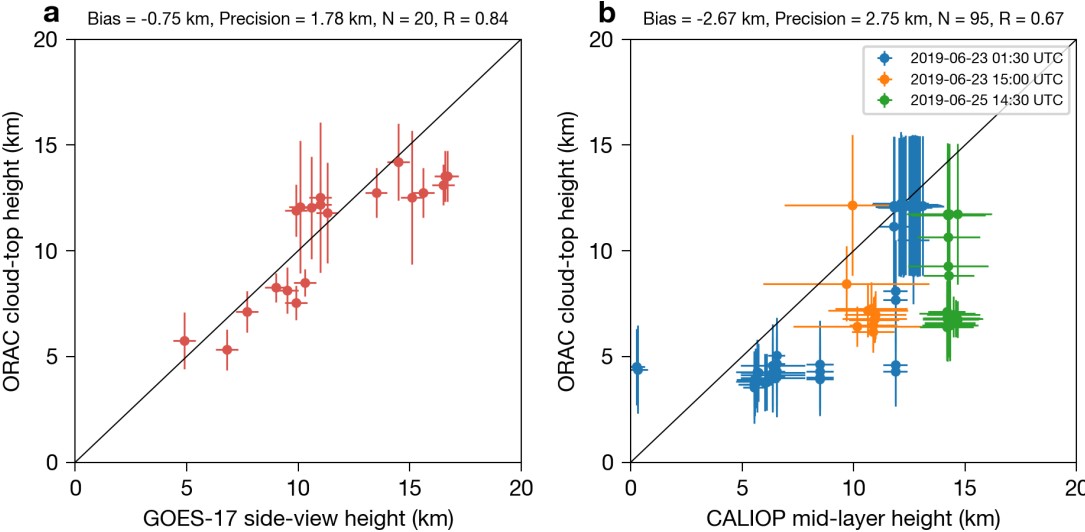

**Figure 9.** (a) Validation results for the ORAC height retrievals compared to the height validation data derived from GOES-17 (side-view heights). (b) Same as (a) but for CALIOP. The heights for CALIOP represent mid-layer heights for the top layer detected by the MLay product (with the error bars representing the top and base heights).

stratospheric volcanic aerosol was too optically thin for the OE to determine the correct height based on the thermal radiances measured by the window channels. Additionally, if the ORAC height retrievals are underestimated, the parallax shift will also be underestimated (for the same satellite zenith angle), leading to collocation errors between AHI and CALIOP. The cloud-top height and associated uncertainty for retrievals in the stratosphere did not deviate significantly from their *a priori* values. This result means that the measurements had little influence on the stratospheric height retrievals. Although the *a priori* height in the stratosphere (200 hPa) was chosen based on CALIOP observations at the beginning of the eruption, this result highlights both the difficulty in determining height in an isothermal lower-stratosphere and the value in setting a representative *a priori* in such conditions.

### 4.5 Importance of the 13.3 μm channel

The Raikoke case study raises several challenges for reliably retrieving ash cloud-top heights in the troposphere and stratosphere from thermal-only satellite measurements. In particular, there are two main issues: 1) the double solution due to the inversion of temperature at the tropopause and 2) an isothermal lower stratosphere. We addressed these issues by selecting the lowest cost from a set of FM configurations that used *a priori* pressures in the troposphere and stratosphere (described in Sect. 3.2.3). To explore the robustness of this method, we ran the five FM configurations again, but removed $T_{13}$ from the measurement vector. We found that when $T_{13}$ was not included, the ability to select the correct retrieval solution was lost.

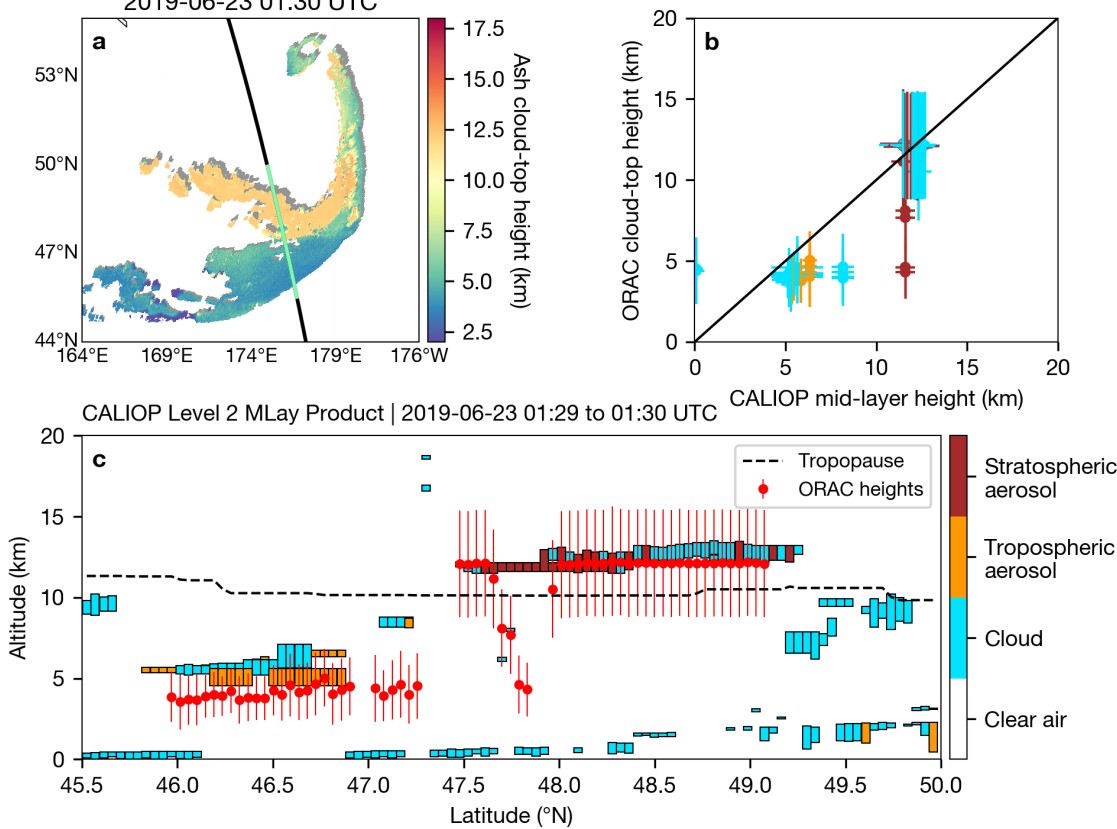

**Figure 10.** (a) Parallax-corrected ORAC ash cloud-top height retrievals at 1:30 UTC on 23 June 2019. Grey shaded region indicates parallax shift. The black line indicates the CALIOP ground track with green indicating the section plotted in (c). (b) Correlation between ORAC ash cloud-top heights and CALIOP mid-layer heights. The colours of the data points represent the feature type identified by the CALIOP vertical feature mask (blue for cloud, orange for tropospheric aerosol and brown for stratospheric aerosol). (c) CALIOP level 2 MLay feature classification flags with collocated ORAC heights over-plotted.

Essentially, for retrievals without the $T_{13}$ channel, the cost for tropospheric and stratospheric height retrievals was very similar, but the stratospheric height solutions always returned the lowest cost.

Given that the information used to retrieve height comes from the thermal channels supplied to the measurement vector, the retrieval is dependent on their respective weighting functions. Figures 11(a) and (b) show the transmittance profiles and weighting functions for a clear atmosphere for each of the channels used in the measurement vector. Comparing Fig. 11(b) with Fig. 11(d) shows that $T_{12}$ is most affected by water vapour in the lower troposphere ($p > 500$ hPa) followed by $T_{11}$ and $T_{13}$, with $T_{10}$ being the least affected by water vapour. The $T_{10}$, $T_{11}$ and $T_{12}$ channel weighting functions go to zero at $\sim 300$ hPa while the $T_{13}$ band follows the decrease in the temperature profile (Fig. 11(c)) and remains non-zero up to $\sim 10$ hPa. The $T_{13}$ weighting function follows the temperature profile because it is sensitive to $CO_2$ absorption which is well-mixed in the

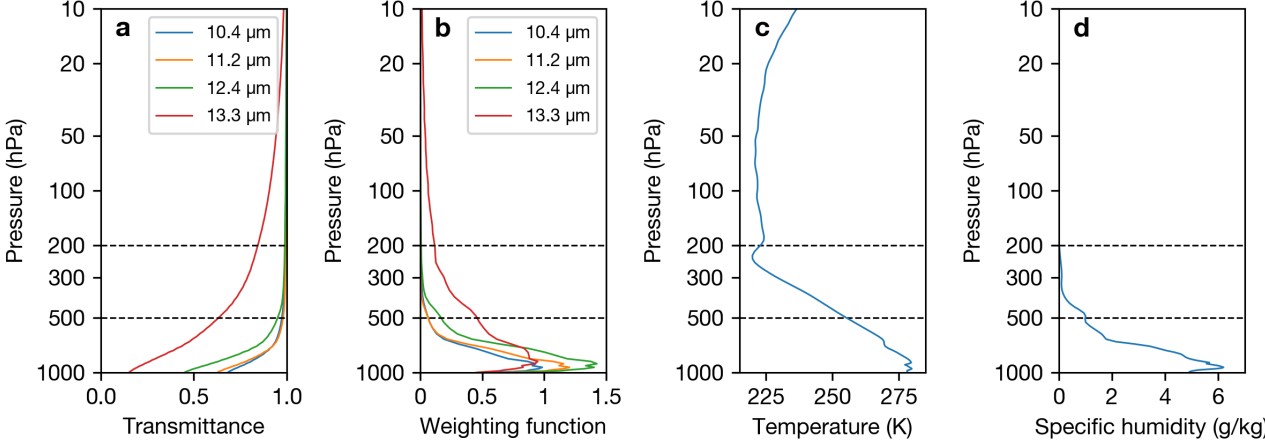

**Figure 11.** (a) Clear-sky transmittance profiles for the $T_{10}$, $T_{11}$, $T_{12}$ and $T_{13}$ AHI channels taken from the ORAC/RTTOV pre-processor output at 18:00 UTC on 22 June 2019. (b) Same as (a) but for the weighting functions. (c) ERA5 temperature profile at the same location as the clear-sky transmittance profiles. (d) Same as (c) but for specific humidity.

atmosphere. Crucially, this variation in the weighing function in the $T_{13}$ channel from 300 hPa to 100 hPa (and above) is what allows the retrieval to distinguish between a cloud layer placed in the troposphere vs the stratosphere based on cost. Without information from the $T_{13}$ channel, there is very little difference in TOA radiance for the simulated $T_{10}$, $T_{11}$ and $T_{12}$ channels for a cloud placed at 500 hPa (troposphere) vs 200 hPa (stratosphere) and so information from the $T_{13}$ is key for distinguishing between these cases. The weighting functions also show that because the lower stratosphere is isothermal, the difference in the $T_{13}$ TOA radiance for a cloud layer placed at 200 hPa, 100 hPa and 50 hPa would be very small, meaning that for a given measurement, the cost would be very similar in each case. This also explains why the retrieved heights in the stratosphere do not deviate significantly from the chosen *a priori* at 200 hPa (Fig. 10(c)).

### 4.6 Importance of the 10.4 μm channel

Figure 12 illustrates the importance of including the 10.4 μm channel in the measurement vector for volcanic ash retrievals using thermal-only channels. By comparing Fig. 12(a) and (b) with (c) one can see that the $T_{11}$-$T_{12}$ and $T_{10}$-$T_{11}$ BTDs are sensitive to different effective radii sizes. The ORAC effective radius retrievals show that small particles are prevalent in the northern part of the volcanic cloud while larger particles are prevalent in the southern part. Inspection of Fig. 12(d) shows that if we were to use a measurement vector that does not include the $T_{10}$ channel then the solution space would be restricted to effective particle sizes larger than ∼2 μm and smaller than ∼7 μm for this particular scene. In addition, without the $T_{10}$ channel, effective radii solutions would be found for particles in the northern part of the ash cloud but would be comparatively larger, leading to higher estimates of the total mass. On the other hand, Fig. 12(e) shows that if $T_{12}$ was left out of the measurement vector, and $T_{10}$ was included, then the retrieval would be sensitive to the smaller particles (for this scene ∼1–

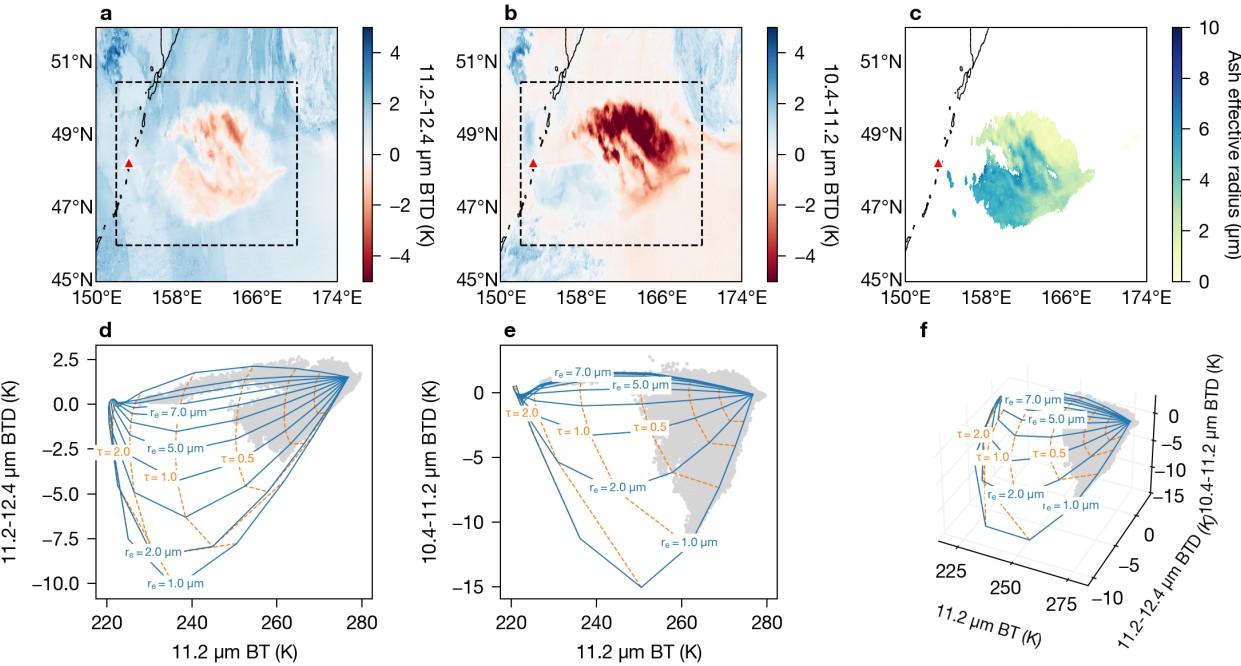

**Figure 12.** (a) Brightness temperature difference between $T_{11}$ and $T_{12}$ at 12:00 UTC on 22 June 2019. Red triangle indicates location of Raikoke. (b) Same as (a) but for a difference between $T_{10}$ and $T_{11}$. (c) ORAC effective radius retrieval. (d) Two-dimensional ORAC look-up table for the complex refractive index of the 2010 Eyjafjallajökull ash (Reed et al., 2018) plotted in the $T_{11}$ and $T_{11}$-$T_{12}$ solution space. Grey data points correspond to pixels within the dashed line, bounding boxes annotated on (a). Lines of constant effective radius are plotted as blue solid lines and lines of constant optical depth (at 550 nm) are plotted as dashed orange lines. (e) same as (d) but for the $T_{11}$ and $T_{10}$-$T_{11}$ solution space. Grey data points correspond to bounding box on (b). (f) Same as (d) and (e) but for the three-dimensional, $T_{11}$, $T_{11}$-$T_{12}$ and $T_{10}$-$T_{11}$ solution space.

2 μm), but restricted to particle sizes smaller than ∼5 μm. In addition, a negative $T_{10}$-$T_{11}$ BTD would not detect the southern part of the ash cloud (Fig. 12(b)). Therefore, by including $T_{10}$, $T_{11}$ and $T_{12}$ in the measurement vector (Fig. 12(f); Supplement Animation 1) we are able to exploit the particle size information in both the $T_{11}$-$T_{12}$ and $T_{10}$-$T_{11}$ BTDs to optimally retrieve a wider range of effective radii sizes than existing retrieval algorithms that only exploit two channels for particle size information (e.g. Corradini et al., 2008; Prata and Prata, 2012; Francis et al., 2012; Pavolonis et al., 2013).

To understand why the combination of the 10.4, 11.2 and 12.4 μm leads to the ability to retrieve a wider range of particle sizes compared to two-channel techniques it is instructive to consider the heuristic model proposed by Prata and Grant (2001) that explains the relationship between the volume extinction coefficient, effective radius and the BTD for a two-channel retrieval. Essentially, the particle size information is captured by the ratio ($\beta$) of volume extinction coefficients at two different wavelengths ($k_1$ and $k_2$) within the thermal infrared window ($\beta = k_2/k_1$). In general, when $\beta > 1$, the BTD is positive indicating ice or water. If $\beta = 1$, the BTD is 0 and we have no information on particle size and if $\beta < 1$, the BTD is negative and

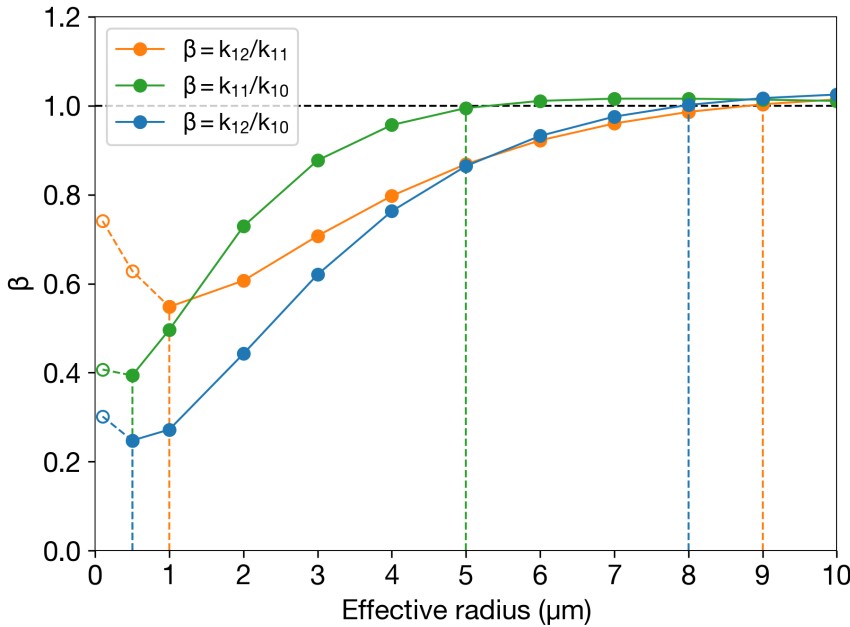

**Figure 13.** Ratios of volume extinction coefficients for different channel combinations as a function of effective radius. Volume extinction coefficients for the 10.4, 11.2 and 12.4 are symbolised as $k_{10}$, $k_{11}$ and $k_{12}$. The dashed lines with open circles indicate where the retrieval of effective radius becomes ambiguous.

we expect volcanic ash particles. Figure 13 shows how $\beta$ varies with effective radius for different channel combinations for the LUTs used in the present study (generated assuming a lognormal size distribution, spherical particles and Eyjafjallajökull ash complex refractive index). The Mie calculations show that the extinction coefficient ratio of the 11.2 and 12.4 μm channels contains information on particles sizes from 0.1–9 μm effective radius; however, there is ambiguity (multiple solutions) once the effective radius reaches 1 μm. In other words, the BTD can be the same for effective radii in the range from 0–1 μm and ~1–3.5 μm. This is not the case for the 10.4 and 11.2 μm combination where $\beta$ continues to decrease as the effective radius reaches 0.5 μm before increasing again. Similar behaviour is seen for the 10.4 and 12.4 channel combination. In addition, the channel combination of 10.4 and 11.2 μm will only return negative BTDs ($\beta < 1$) for effective radii up to 5 μm. This prediction of the Mie theory explains why only the northern part of the Raikoke plume is detected by the 10.4-11.2 BTD, whereas the whole plume is detected by the 11.2-12.4 BTD (compare Figs.12(a) and (b)). Overall the combination of 10.4, 11.2 and 12.4 allows for an unambiguous retrieval of effective radius from 0.5–9 μm.

## 5  Conclusions

In this study we have presented uncertainty-bounded estimates of volcanic ash cloud-top height, optical depth at 550 nm, effective radius and mass loading for the June 2019 Raikoke eruption. We found the Raikoke eruption injected $0.73 \pm 0.40\,\mathrm{Tg}$ of very fine ash into the troposphere and stratosphere. After reaching its maximum, $\sim 90\,\%$ of the total mass was removed from the atmosphere over 48 h (*e*-folding time of 20 h) with 0.10 Tg detectable in the atmosphere for at least four days, corresponding to median ash concentrations of $\sim 0.2$–$2\,\mathrm{mg\,m^{-3}}$ (depending on the geometric thickness assumed). The distal fine ash mass fraction was estimated to be $0.73 \pm 0.62\,\%$ based on the total very fine ash mass retrieved and the ORAC cloud-top heights converted to mass eruption rates based on the Mastin relationship. Our analysis shows that the Raikoke source term is highly complex, meaning that eruption source parameters (i.e. injection height, duration and mass eruption rate) must be carefully considered when attempting to model this eruption. For example, if a continuous eruption source with a constant maximum height is assumed then it is likely that the total mass will be overestimated due to the numerous pauses and variations in height during this eruption sequence (e.g. Bruckert et al., 2022). Even if the duration of the eruptions are accurately captured, underestimates or overestimates could occur if the plume height is not allowed to vary between the troposphere and stratosphere with time.

The ORAC algorithm represents several advances in thermal IR-based ash retrieval algorithms applied to geostationary satellite measurements. Advances include a better characterisation of measurement noise that is allowed to vary with the measured brightness temperature, the ability to distinguish between heights in the troposphere and stratosphere based on cost (with the inclusion of the $T_{13}$ channel), the retrieval of a wider range of effective radii sizes (with the inclusion of the $T_{10}$ channel) and accounting for underlying meteorological clouds in the FM. The ash cloud-top height retrievals representing the near-source plume showed good agreement ($R = 0.84$) when compared against GOES-17 side-view height data but showed a notable negative bias (-2.67 km) for the distal ash clouds when compared against CALIOP data. Caution must be exercised when interpreting the ORAC ash cloud-top heights in the stratosphere for the distal ash clouds as the retrieved solutions deviated very little from their *a priori* values due to the isothermal nature of lower stratosphere. One improvement that could be made would be to use additional information, such as from dispersion model simulations, to set tightly constrained *a priori* pressure fields where ash is detected by AHI. This approach would essentially be using information from the wind fields in the stratosphere to overcome a flat cost surface due to the isothermal lower-stratosphere. Other improvements include using visible channels in the measurement vector to determine optical depth and effective radius in regions of the plume that are opaque to thermal IR measurements (AHI bands centred near 0.51, 0.64, 0.86, 1.6, 2.3 and 3.9 μm channels would be suitable; Prata and Grant, 2001; McGarragh et al., 2015), retrieving other multi-layer scenarios (e.g. ice above ash, ash above ash etc) and improve the ash detection flag using machine learning (Picchiani et al., 2011; Gray and Bennartz, 2015; Piontek et al., 2021a).

The ORAC retrievals provided here could either be used as a validation dataset for dispersion model simulations or incorporated into data assimilation schemes that require uncertainties at the grid/pixel level (e.g. Mingari et al., 2022). Incorporation of these retrievals into such schemes could be used to develop quantitative now-casting/forecasting products that will aid VAACs in providing advice to airlines about quantitative ash concentrations in the future. In terms of implementing the retrieval scheme

operationally, we have shown that one could use default *a priori* pressure settings in the troposphere and stratosphere and then select the run configuration with the lowest cost on a per-pixel basis to identify whether or not ash is present in the stratosphere. Height estimates could then be refined as information from independent sources becomes available (e.g. from lidar, geostationary parallax and side-view heights).

*Code and data availability.* The ORAC code is open source and available from Github (https://github.com/ORAC-CC/orac). The ORAC 10-minute time-series of plume-top heights and 1-h retrievals (all state variables and ash flag) out to 7-days after eruption are available upon request from the corresponding author.

## Appendix A:  Error propagation in the Mastin equation

The relationship between the volumetric flow rate and the plume height has been developed over many years (Morton et al., 1956; Wilson et al., 1978; Settle, 1978; Bursik et al., 1992; Sparks et al., 1997; Mastin et al., 2009):

$$H = a\dot{V}^b, \tag{A1}$$

where $H$ is the plume height above vent level (m), $\dot{V}$ is the volumetric flow rate ($\text{m}^3/\text{s}$) and $a$ and $b$ are free parameters determined from an empirical fit. Equation A1 can be rearranged to solve for the mass eruption rate, $\dot{M}$ (kg/s), as follows:

$$\dot{M} = \rho_d \left( \frac{H}{a} \right)^{1/b}, \tag{A2}$$

where $\rho_d$ is the 'Dense Rock Equivalent' density of tephra (all particle sizes erupted from the volcano), which is typically assumed to be $2500 \, \text{kg} \, \text{m}^{-3}$ (Mastin et al., 2009; Dioguardi et al., 2020). Note this density is distinct from the very fine ash particle density, $\rho$, which we assumed was $2300 \pm 300 \, \text{kgm}^{-3}$ in the present study. If we assume all variables in Eq. A2 are independent then we can compute the uncertainty in $\dot{M}$ from the partial derivatives as follows:

$$\frac{\partial \dot{M}}{\partial \rho_d} = \left( \frac{H}{a} \right)^{1/b} \tag{A3}$$

$$\frac{\partial \dot{M}}{\partial H} = \frac{1}{bH} \cdot \rho_d \left( \frac{H}{a} \right)^{1/b} \tag{A4}$$

$$\frac{\partial \dot{M}}{\partial a} = -\frac{1}{ba} \cdot \rho_d \left( \frac{H}{a} \right)^{1/b} \tag{A5}$$

$$\frac{\partial \dot{M}}{\partial b} = -\frac{\ln(H/a)}{b^2} \cdot \rho_d \left( \frac{H}{a} \right)^{1/b} \tag{A6}$$

The uncertainty in $\dot{M}$ is then

$$(\alpha_{\dot{M}})^2 = \left(\frac{\partial \dot{M}}{\partial \rho_d}\right)^2 (\alpha_{\rho_d})^2 + \left(\frac{\partial \dot{M}}{\partial H}\right)^2 (\alpha_H)^2 + \left(\frac{\partial \dot{M}}{\partial a}\right)^2 (\alpha_a)^2 + \left(\frac{\partial \dot{M}}{\partial b}\right)^2 (\alpha_b)^2, \tag{A7}$$

which rearranges to

$$\left(\frac{\alpha_{\dot{M}}}{\dot{M}}\right)^2 = \left(\frac{\alpha_{\rho_d}}{\rho_d}\right)^2 + \left(\frac{1}{b} \cdot \frac{\alpha_H}{H}\right)^2 + \left(\frac{1}{b} \cdot \frac{\alpha_a}{a}\right)^2 + \left(\frac{\ln(H/a)}{b} \cdot \frac{\alpha_b}{b}\right)^2$$

$$= \left(\frac{\alpha_{\rho_d}}{\rho_d}\right)^2 + \frac{1}{b^2}\left[\left(\frac{\alpha_H}{H}\right)^2 + \left(\frac{\alpha_a}{a}\right)^2 + \ln^2(H/a) \cdot \left(\frac{\alpha_b}{b}\right)^2\right]. \tag{A8}$$

$$\alpha_{\dot{M}} = \pm \dot{M} \cdot \sqrt{\left(\frac{\alpha_{\rho_d}}{\rho_d}\right)^2 + \frac{1}{b^2}\left[\left(\frac{\alpha_H}{H}\right)^2 + \left(\frac{\alpha_a}{a}\right)^2 + \ln^2(H/a) \cdot \left(\frac{\alpha_b}{b}\right)^2\right]}. \tag{A9}$$

From Eq. A8 we can see that the relative uncertainty in $\rho_d$ is proportional to the relative uncertainty in $\dot{M}$ whereas an increase in the relative uncertainty in $H$ or $a$ will result in a factor $1/b$ increase in the relative uncertainty in $\dot{M}$. In addition, as the relative uncertainty in $b$ increases, the relative uncertainty in $\dot{M}$ will be an exponential function of $H$ due to the $\ln(H/a)/b$ term and will thus dominate the uncertainty budget for large $H$ (e.g. $> 5.5$ km; for $b = 0.241$).

To compute the total mass, $M_T$, and its associated uncertainty, $\alpha_{M_T}$, we must integrate $\dot{M}$ with respect to time, $t$:

$$M_T = \int_{t_0}^{t_n} \dot{M}(t) dt. \tag{A10}$$

The above integral can be approximated as a sum of mass eruption rates multiplied by discrete time periods (i.e. the temporal resolution of the satellite), $\Delta t$, so that

$$M_T \approx \sum_{i=0}^{n} \dot{M}_i \Delta t = \sum_{i=0}^{n} M_i, \tag{A11}$$

where $n$ is the total number of observation times, $\dot{M}_i$ is the mass eruption rate at time, $i$, and $M_i$ is the total mass at time, $i$, assuming constant $\dot{M}_i$ over $\Delta t$ (note that for AHI, $\Delta t = 600$ s). The uncertainty in the total mass can therefore be written as

$$(\alpha_{M_T})^2 = (\alpha_{M_1})^2 + (\alpha_{M_2})^2 + \cdots + (\alpha_{M_n})^2 \tag{A12}$$

$$\alpha_{M_T} = \pm \Delta t \sqrt{(\alpha_{\dot{M}_1})^2 + (\alpha_{\dot{M}_2})^2 + \cdots + (\alpha_{\dot{M}_n})^2}, \tag{A13}$$

where we assume errors across each time step are uncorrelated. To evaluate $\dot{M}$ and $\alpha_{\dot{M}}$ at all observation times using Eqs. A2 and A9, we use the ORAC height time-series shown in Fig. 5. Note that $H$ is determined from the ORAC-retrieved height, $h_c$, via $H = h_c - h_v$, where $h_v$ is the vent height of the Raikoke volcano (0.551 km above sea level). For $\rho_d$, $a$ and $b$, we use the values provided in Mastin et al. (2009), which are $2500$ kg m$^{-3}$, $2.00$ and $0.241$, respectively. Uncertainty in $H$ is taken from the ORAC retrievals. Uncertainty in $\rho_d$ is not provided in Mastin et al. (2009) and so we conservatively assumed a relative uncertainty on $\rho_d$ of 50%. This value may seem high, but when propagated through Eqs. A9 and A13 the difference between a

relative uncertainty of 10% and 50% converts to a difference in absolute uncertainty in the total mass of $\sim$1 Tg. Uncertainty
associated with the $a$ and $b$ parameters is related to the sample size, eruption type and errors in plume height and volumetric
flow rate taken from the cases used in the empirical fit. Based on the range of values reported and discussed in the literature
(e.g. Bursik et al., 1992; Sparks et al., 1997; Mastin et al., 2009) and noting the theoretical finding that $\dot{M}$ is related to $H$ to the
fourth power (i.e. $b = 0.25$; Morton et al., 1956), we assumed relative uncertainties on $a$ and $b$ of 90% and 20%, respectively.

Finally, the distal fine ash mass fraction, $m_f$, is computed as

$$m_f = \frac{m_T}{M_T},\tag{A14}$$

where $m_T$ is the maximum total fine ash mass retrieved determined from ORAC. The associated uncertainty in $m_f$ is

$$\alpha_{m_f} = \pm m_f \cdot \sqrt{\left(\frac{\alpha_{m_T}}{m_T}\right)^2 + \left(\frac{\alpha_{M_T}}{M_T}\right)^2},\tag{A15}$$

where $\alpha_{m_T}$ is the uncertainty associated with the maximum total mass of very fine ash determined from ORAC.

*Author contributions.* ATP, RGG and IAT developed the ideas for the manuscript. ATP led the writing of the manuscript, ran the satellite
retrievals and conducted the data analysis. RGG, ACP, SRP, ATP and CP are developers of the ORAC software and made improvements to
the code for the purposes of this study. All authors contributed to data interpretation and writing of the manuscript.

*Competing interests.* The authors declare no competing interests.

*Acknowledgements.* RGG and IAT were supported by the NERC Centre for Observation and Modelling of Earthquakes, Volcanoes, and
Tectonics (COMET). This study was funded as part of NERC's support of the National Centre for Earth Observation, contract number
PR140015. ATP and RGG acknowledge funding from the NERC R4AsH project NE/S003843/1. IAT and RGG acknowledge funding from
the NERC V-Plus project NE/S004025/1. We thank two anonymous reviewers for their detailed and thoughtful reviews both of which helped
to significantly improve the manuscript.

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
