# Peer review of "Uncertainty-bounded estimates of ash cloud properties using the ORAC algorithm: Application to the 2019 Raikoke eruption"

_Atmospheric Measurement Techniques, 2022_

## Referee Comment (RC2)

General comments

In this paper, the authors apply the optimal estimation ORAC algorithm to retrieve the macro- and microphysical properties of the 2019 Raikoke volcanic ash clouds using geostationary thermal infrared imagery. ORAC is a well-established algorithm with several recent publications describing the method and its validation, mostly in meteorological clouds. The application of a flexible and, importantly, open-source code to volcanic ash retrievals is a welcome addition to the literature. The authors highlight the potential advantages of their approach: formal error estimates, handling of multilayer situations, improved effective radius estimation by inclusion of an additional IR channel, and better treatment of ambiguities in height retrieval.

The paper is well-written and easy to follow, the figures are of good quality. My main criticism is that the improvement in height retrievals seems a bit overstated. Plume height is the only parameter the authors attempt to validate in the current study against lidar and geometric height estimates. The presented comparison, however, indicates that ORAC has limited skill in retrieving stratospheric plume heights, at least in the current case characterized by a nearly isothermal lower stratosphere. Nevertheless, I recommend the manuscript for publication after minor revisions. Below is a list of questions and suggestions the authors might want to consider for improvement.

Specific comments

- Lines 143-144: Can you explain, in a sentence or two, the main differences between the ORAC cloud and aerosol FM that made you choose the cloud model for volcanic ash?

- Line 190: You assume spherical particles, although volcanic ash clouds often contain non-spherical particles. For example, the MISR Active Aerosol Plume-Height project (V. Flower, R. Kahn, J. Limbacher / NASA GSFC) retrieved aerosol properties for a low-altitude part of the Raikoke plume on 23 June 2019 from multiangle observations, which indicate a high fraction (>0.6) of non-spherical particles.

  Is ORAC able to handle non-spherical particles in general? Can you speculate about the retrieval errors caused by assuming spherical particles for non-spherical ones?

  The MISR results are available at https://appliedsciences.nasa.gov/our-impact/news/misr-plume-heights-and-aerosol-characteristics-2019-raikoke-eruption, although the linked Powerpoint file seems corrupted at the moment.

- Lines 285-286 and Eq. 10: The definition of ash mass loading is indeed the same as that of cloud liquid water path (LWP) and assumes no vertical variation in cloud/ash particle size. For marine stratocumulus clouds, however, effective radius often increases linearly from cloud base to top. Because the effective radius derived from VIS-NIR bispectral observations is representative of the cloud top, the assumed vertical homogeneity results in an overestimated LWP. For such boundary layer clouds, an 'adiabatic' model has been proposed, which leads to a proportionality factor of 10/9 instead of 4/3 in Eq. 10. This adiabatic correction reduces the VIS-NIR LWP by 17%, which agrees better with microwave LWP estimates for marine Sc.

Your estimate of the Raikoke ash cloud geometric thickness is 1.04±0.56 km, which is larger than the ~300 m typical thickness of marine Sc. Thus, vertical variations in ash particle effective radius can also introduce biases in ash mass loading when vertical homogeneity is assumed.

Can you comment on which part of the ash cloud (top, middle, bottom) your IR-based effective radius estimates are representative of? Do you expect vertical variations in effective radius—perhaps a decrease from base to top, in contrast to marine Sc—and how would these bias your ash mass load estimates?

- Line 344: "***there is no information on particle size for opaque plumes***". Perhaps you could note this limitation earlier, when describing the method in section 3.

- Lines 359-360: You state that "***The ORAC heights are, however, an improvement to simply matching a brightness temperature to a NWP profile***…"; but, you don't demonstrate this. Could you perhaps plot the height or height range derived by matching the minimum plume BT within the black circle in Fig. 3a to the ERA5 profile?

- Lines 360-362: How big are the differences in the 5 cost values? Are these differences significant?

- Lines 389-391: You suggest that the increase in effective radius is due to a retrieval artefact at large satellite view zenith angle (VZA). Cloud drop effective radius retrieved by the plane-parallel VIS-NIR bispectral method has been found to increase at large VZA due to 3D effects (shadowing, illumination) affecting the VIS reflectances (Maddux et al., 2010, 10.1175/2010JTECHA1432.1; Grosvenor and Wood, 2014, 10.5194/acpd-14-303-2014; Horvath et al. 2014, 10.1002/2013JD021355).

  How does VZA affect your IR measurements that could explain this retrieval artefact? Is it differential limb cooling in the various channels due to increased absorption by $CO_2$ and/or water vapor? Can you explain it by the topology (shape of constant effective radius curves) of the ORAC look-up tables plotted in Fig. 10?

- Lines 459-460: You state that there is generally good agreement between the ORAC and validation heights. However, Fig. 7 suggests that the agreement between ORAC and GOES-17 is better in the troposphere, while the skill of ORAC in the stratosphere seems limited. GOES-17 indicates heights between 10-16km, while ORAC retrieves near constant heights around 12-13km. You state elsewhere that the IR measurements had little influence on the retrieved stratospheric heights, which were close to the chosen a priori near 12km. Perhaps a more nuanced statement on the height comparison is needed here.

- Lines 479-482: Purely as a suggestion for future study, I recommend analysing ORAC retrievals for the 2021 La Soufriere eruptions, where the tropical temperature profile shows a sharp inversion at the tropopause and distinct temperature increase in the stratosphere. ORAC stratospheric height retrievals might have better skill in a tropical case than in the current mid-latitude case characterized by an isothermal lower stratosphere. CALIOP and GOES-17 height estimates are available for La Soufriere too.

- Lines 529-534: Bruckert et al. (2022, 10.5194/acp-22-3535-2022) in the same special issue also show that resolving the individual phases of the 2019 Raikoke eruptions

improves the total ash burden forecast and reduces ash mass overestimation, compared to using continuous and constant eruption source parameters.

Technical corrections

- Line 49: "in satellite-based**,** ash cloud retrieval" -> remove the comma

- Line 173: "interpola**R**ted" -> "interpolated"

- Line 183: Is $\lambda$ the wavelength?

- Line 258: "problem of **a** multiple solutions" -> remove "a"

- Line 265: "a tropospheric**,** a priori ash layer pressure" -> remove the comma

- Line 268: "which is a typical **of** value for ash" -> remove "of"

- Line 417: "side-view**,** times-series" -> remove the comma

- Line 449: "were ~0.3–0.7 km according to" -> "…were ~0.3–0.7 km **thick**…"

- Line 455: "would **be** fall below" -> remove "be" or "fall"

- Line 501: "**their** is very little" -> "there"

- Fig. 8a: I presume, the black line is the CALIOP ground track with green indicating the section plotted in panel c.

- Line 504: "for a cloud layer**s**" -> "layer"

- Line 602: "converts to a **different** in absolute" -> "difference"

---

## Author Comment (AC1)

**Uncertainty-bounded estimates of ash cloud properties using the ORAC algorithm: Application to the 2019 Raikoke eruption**

Andrew T. Prata, Roy G. Grainger, Isabelle A. Taylor, Adam C. Povey, Simon R. Proud and Caroline A. Poulsen

We thank both reviewers for their detailed and thoughtful reviews both of which helped to significantly improve the manuscript. Please find our responses (in blue) to review comments (in black). We have also listed some technical corrections that we found during the review process at the end of this document and have made some very minor edits to the text to improve readability (which can be seen in the track changes document).

**Response to RC1**

**Specific comments**

Line 63: There are newer contributions to the question of ash composition variability and its influence on the radiative transfer, which you might want to mention here, e.g., Prata et al. (2019, 10.1029/2018JD028679), Deguine et al. (2020, 10.1364/AO.59.000884), Piontek et al. (2021, 10.1016/j.jvolgeores.2021.107174)

Thanks. We've added these references in our revision.

Line 144: Why do you use the cloud instead of the aerosol implementation of the radiative transfer forward model for volcanic ash?

In ORAC, clouds are considered as a geometrically infinitesimal layer within the atmosphere. RTTOV is used to produce profiles of atmospheric transmission/reflectance from which the layer height is extrapolated. Aerosols are considered as a continuum. RTTOV is not run in this case as the multiple scattering is considered within the DISORT calculations that generate the LUTs. The current implementation for aerosols does not consider IR wavelengths. We decided that the forward model representing cloud would be more suitable for volcanic ash clouds as they are often observed as well-bounded features in the vertical (see numerous examples of CALIPSO observations) rather than a well-mixed continuum distributed over a vertical region of the atmosphere. This choice has a practical advantage as well as we wanted to run consistent retrievals over day and night (the aerosol model can currently only be used during day). We've added a brief description of the difference between the aerosol and cloud models in our revision.

Lines 145-146: Why do you focus on the thermal channels in the measurement vector?

The Raikoke ash clouds dispersed into the atmosphere over several days and nights. The main reason for selecting thermal channels was to ensure consistency over day and night in the retrievals. We've now explicitly stated this in the revised manuscript.

Line 186: It would be interesting to know which bulk silica contents the three used volcanic ash samples had, as well as which silica content the Raikoke ash had (if corresponding measurements are available in the literature). You found that the Eyjafjallajökull ash refractive index lead to the best retrieval results. Do Eyjafjallajökull ash and Raikoke ash have similar silica contents or are they otherwise comparable in composition?

The bulk silica contents for Eyja, Spurr and Chaiten are 58.85, 55.99 and 74.90 wt%. Prata et al. (2019) provide a summary table of bulk silica contents in the Oxford ARIA database. At the time of writing our initial submission we were unaware of any bulk silica contents for the Raikoke ash. However, a recent study by Smirnov et al. (2021) provides bulk silica contents for samples representing the 21–26 June 2019 eruption. For "bulk compositions of air fall and PDC ashes" the bulk silica contents are between 50–55 wt%. For "glass compositions of shards from air fall ash" bulk silica contents are mostly between 57–63 wt% (see their TAS diagram in Fig. 5 top panel). Therefore the bulk composition of 58.85 wt% (used in our study) appears to be consistent with the glass shards of airfall ash reported in Smirnov et al. (2021). We have added this discussion to the revised manuscript.

Lines 214-215: Why do you not retrieve the cloud fraction f as well? The assumption of f being equal to one is certainly true well inside the volcanic ash cloud, but what about the edges of the ash cloud?

Experience using ORAC to retrieve cloud fraction for meteorological clouds has shown that there is a compensating effect if optical depth and cloud fraction are simultaneously retrieved. Essentially the optical depth increases as the cloud fraction is reduced (and vice-versa). One approach to addressing this issue is to tightly constrain cloud fraction if it can be estimated from higher resolution data. Watts et al. (1998) and Poulsen et al. (2012) discuss this in more detail. Investigating this issue would be beyond the scope of the present study and so for now we assume a cloud fraction of 1 and account for uncertainty due to cloud inhomogeneity (i.e. broken cloud or pixels at cloud edges), in addition to errors due to the plane-parallel cloud assumption, as a forward model error (0.50 K for thermal channels), which stems from the work of Watts et al. (1998). We have added this justification to the revised manuscript.

Line 271: Is there a specific reason for setting the a priori uncertainties of $\tau$ and $r_e$ to $10^8$, or is this just an arbitrary high number?

The $10^8$ value is arbitrary but is set to a large number to ensure no influence of the *a priori* uncertainty on the retrieved values of $\tau$ and $r_e$ (i.e. completely flat probability distribution over the whole range of possible parameter values). We added this note to the revised manuscript.

Lines 276-277: It would be interesting to actually see the impact of considering five different setups with single and multi layer cloud structures. Maybe you can quantify the retrieval improvement of your setup compared to just considering a single volcanic ash layer without underlying water clouds, or maybe you

[Figure]

Figure 1: (a)–(e) Measurement cost at solution for each of the forward model configurations (annotated above each subplot) used in the present study. (f) Combined cost for each configuration (i.e. minimum cost per pixel out of the 5 configurations considered). (g) 11.2 μm brightness temperature. (h) Forward model flag (i.e. forward model configuration that resulted in the lowest cost per pixel). Natural colour composite is plotted beneath for context.

can show an example plot indicating which retrieval configuration was used for which pixel?

Thank you for this comment. This was a great suggestion. We have now added a new figure to the revised manuscript that shows each cost map, the combined cost map (where the minimum cost for each configuration is selected on a per-pixel basis) and a 'classification map' where we have coloured each pixel according to the retrieval configuration that resulted in the lowest cost. We have also plotted the $T_{11}$ brightness temperature and a natural colour composite to provide contextual information that shows that the selected forward model configuration is reasonable (e.g. the stratospheric ash over 500 hPa water cloud is selected for ash overlying the cold cloud associated with the cyclone). We have added text to discuss this new figure (see Fig. 1) in our revision.

Line 285: Can you give a motivation for your choice of the ash particle density of 2300 kg m$^{-3}$?

The main reason for using the particle density of 2300 kg m$^{-3}$ was to be consistent with what is used in the NAME dispersion model as an earlier version of the retrievals presented here were used for comparison to the model in Harvey et al. (2022). This ash density was determined for operational use and is therefore a representative average value (Witham et al., 2019). We have accounted for uncertainty in this value by allowing for an absolute uncertainty of 300 kg m$^{-3}$. We have added this justification to the revised manuscript.

[Figure]

Figure 2: (a) Mass loading retrieval before gap filling processing step. (b) Mass loading after gap filling processing step. Note this gap filling processing step occurs before parallax correction.

Lines 301-303: Can you give a motivation for the thresholds? Were they chosen upon manual inspection of the satellite data?

The thresholds were chosen based on manual inspection of the data. We have now stated this explicitly in the revised manuscript.

Lines 314-320: How many gaps did you observe? Were they just single pixels? Can you quantify the amount of ash-contaminated pixels in the final data set that resulted from gap filling compared to the amount that directly resulted from the optimal estimation retrieval? Again it might be nice to see an example image to get a better impression on the importance/significance of this processing step.

The number of gaps varies with time. In general the number of gaps increased as the total number of ash-contaminated pixels increased. At certain times, the fraction of gap filled pixels is not insignificant (reaching as high as ∼34 %). It is important to note, however, that at the time of the maximum total mass of very fine ash (07:00 UTC on 22 June 2019), the fraction of gap filled pixels is ∼7 %; meaning that, regardless of gap filling, the total mass estimate would be within the uncertainty range given. We have added this discussion to the revised manuscript and we have added a new figure (Fig. 2) showing the before and after gap filling processing step.

Line 526: The formulation "exponential decay rate of 20 h" is a bit misunderstandable, as rate rather refers to a change per time. Better give a "half-life time" or "e-folding time" or similar.

Thanks for picking up on this. We have corrected to "e-folding time" in our revision.

Figure 10: Panel (f) is hard to understand, but there is probably no better way to visualize this.

Indeed, we did think about how to represent this. The 3D plot is best viewed as an interactive plot. To present this to the reader we have now added an animation that rotates the 3D plot. It has been added as a Supplementary Animation 1.

Line 538: Can you quantify the expression "wider range of effective radii sizes"?

To understand why the combination of the 10.4, 11.2 and 12.4 µm leads to the ability to retrieve a wider range of particle sizes compared to two-channel techniques it is instructive to consider the heuristic model proposed by Prata and Grant (2001) that explains the relationship between the volume extinction coefficient, effective radius and the BTD for a two-channel retrieval. Essentially, the particle size information is captured by the ratio of volume extinction coefficients ($\beta = k_2/k_1$). In general, when $\beta > 1$, the BTD is positive indicating ice or water. If $\beta = 1$, the BTD is 0 and we have no information on particle size and if $\beta < 1$, the BTD is negative and we expect volcanic ash particles. Figure 3 shows how $\beta$ varies with effective radius for different channel combinations for the LUTs used in the present study (generated assuming a lognormal size distribution, spherical particles and Eyja ash complex refractive index). The Mie calculations show that the extinction coefficient ratio of the 11.2 and 12.4 micron channels contains information on particles sizes from 0.1–9 µm effective radius; however, there is ambiguity (multiple solutions) once the effective radius reaches 1 µm. In other words, the BTD will be the same for effective radii from 0–1 µm and 1–3.5 µm. This is not the case for the 10.4 and 11.2 µm combination where the $\beta$ continues to decrease as the effective radius reaches 0.5 µm before increasing again. Similar behaviour is seen for the 10.4 and 12.4 channel combination. In addition, the channel combination of 10.4 and 11.2 µm will only return negative BTDs ($\beta < 1$) for effective radii up to 5 µm. This prediction of the Mie theory explains why only the northern part of the Raikoke plume is detected by the 10.4-11.2 BTD, whereas the whole plume is detected by the 11.2-12.4 BTD (see Figs. 10(a) and (b) of the original manuscript). Overall the combination of 10.4, 11.2 and 12.4 allows for an unambiguous retrieval of effective radius from 0.5–9 µm. We have explained this in the revised manuscript and accompanied it with the new figure (Fig. 3) that shows the particle size sensitivity to the volume extinction coefficients used in the LUTs for the present study.

Line 545: If possible, can you give recommendations which visible channels to use?

Prata and Grant (2001) suggest that a ratio of visible to NIR reflectances could be used to discriminate ash from water/ice based on the fact that the imaginary refractive index of andesite at 0.63 µm is 5 orders of magnitude larger than that of water and ice. In addition, ash absorbs more strongly than ice and water at 1.61 µm (see their Table 2). Some earlier work from our group on this (i.e. McGarragh et al., 2015) found promising results using VIS/NIR MODIS channels centred near 0.645, 0.858, 0.555, 1.24, 1.64, 2.13, 3.75, 1.10, 1.20, 1.33, 1.36, 1.39 and 1.42 µm. The corresponding VIS/NIR AHI wavelengths are: 0.51, 0.64,

[Figure]

Figure 3: Ratios of volume extinction coefficients for different channel combinations as a function of effective radius.

0.86, 1.6, 2.3 and 3.9 µm. Therefore we suggest these channels would be a good place to start. We have added this discussion in our revised manuscript.

Line 547: Machine learning was already used for the volcanic ash detection flag, e.g., Picchiani et al. (2011, 10.5194/amt-4-2619-2011 ), Gray and Bennartz (2015, 10.5194/amt-8-5089-2015), Piontek et al. (2021, 10.3390/rs13163112)

Thanks. We've added these references to qualify this statement.

**Technical corrections**

Lines 28-29: Change order of a and b in "Pavolonis et al., 2015b, a".

Done.

Lines 43-44: The formulation "absorption properties act in opposite directions across the thermal IR window" is a bit unclear, consider to reformulate it.

Accepted. The new sentence is: "Volcanic ash clouds can be discriminated from water and ice clouds because the absorption of thermal radiation for ash decreases from 10–12 µm while thermal infrared absorption increases from 10–12 µm for water and ice, a property known as the 'reverse absorption' effect."

Line 45: Mention at which wavelength the optical depth is considered, here and

everywhere else in the manuscript. Different retrievals consider different optical depths of volcanic ash, e.g., Piscini et al. (2014, 10.5194/amt-7-4023-2014) consider the aerosol optical depth at 11 µm.

Done.

Line 60: I suggest writing "radius ≤ 15 µm".

Done.

Lines 119-120: I suggest to note the band's central wavelength instead of the band number for better understanding if the reader is unfamiliar with the instrument.

Done.

Line 124: "This correspondS to one week of observations […]"

Done.

Lines 127-128: Add the longitude position of GOES-17 as you did for Himawari-8.

Done.

Line 135: Is the minus sign in "-0.5 km" correct?

Yes. CALIOP can actually penetrate the sea surface.

Line 140: "The Deutscher Wetterdienst (DWD) HAS developed the code […]"

Done.

Line 231: Write out abbreviation "SO2" when first mentioned.

Done.

Table 1: In the a priori column, does the first number correspond to the pressure level of ash and the second to the water cloud? If yes, mention this in the caption.

Yes. Done.

Figure 4: Add description of the blue line in the plots in the corresponding caption.

Done.

Line 382: "e.g. 1.1 ± 7 Tg IN Muser et al., 2020"

Done.

Line 530: Comma: „[…] complex, meaning […]"

Done.

Lines 531-532: The sentence would be better understandable if you move "due to the numerous pauses and variations in height during this eruption sequence" to the end.

Done.

Line 534: Wrong spelling: "stratoSPhere"

Done.

**Response to RC2**

The paper is well-written and easy to follow, the figures are of good quality. My main criticism is that the improvement in height retrievals seems a bit over-stated. Plume height is the only parameter the authors attempt to validate in the current study against lidar and geometric height estimates. The presented comparison, however, indicates that ORAC has limited skill in retrieving strato-spheric plume heights, at least in the current case characterized by a nearly isothermal lower stratosphere. Nevertheless, I recommend the manuscript for publication after minor revisions. Below is a list of questions and suggestions the authors might want to consider for improvement.

We thank the reviewer for these comments. The reviewer is correct that ORAC showed limited skill in retrieving stratospheric heights. The main finding that we were trying to highlight is the ability to distinguish between tropospheric and stratospheric heights (with the use of the 13.3 micron channel). In our revision we have been careful not to overstate the skill that ORAC has in estimating stratospheric heights for the Raikoke case and have highlighted the point that the 13.3 micron channel is important when attempting to distinguish between volcanic ash in the stratosphere and troposphere.

**Specific comments**

Lines 143-144: Can you explain, in a sentence or two, the main differences be-tween the ORAC cloud and aerosol FM that made you choose the cloud model for volcanic ash?

In ORAC, clouds are considered as a geometrically infinitesimal layer within the atmosphere. RTTOV is used to produce profiles of atmospheric transmis-sion/reflectance from which the layer height is extrapolated. Aerosols are con-sidered as a continuum. RTTOV is not run in this case as the multiple scattering

is considered within the DISORT calculations that generate the LUTs. The current implementation for aerosols does not consider IR wavelengths. We decided that the forward model representing cloud would be more suitable for volcanic ash clouds as they are often observed as well-bounded features in the vertical (see numerous examples of CALIPSO observations) rather than a well-mixed continuum disturbed over a vertical region of the atmosphere. This choice has a practical advantage as well as we wanted to run consistent retrievals over day and night (the aerosol model can currently only be used during day). We've added a brief description of the difference between the aerosol and cloud models in our revision.

Line 190: You assume spherical particles, although volcanic ash clouds often contain non-spherical particles. For example, the MISR Active Aerosol Plume-Height project (V. Flower, R. Kahn, J. Limbacher / NASA GSFC) retrieved aerosol properties for a low-altitude part of the Raikoke plume on 23 June 2019 from multiangle observations, which indicate a high fraction (>0.6) of non-spherical particles.

Is ORAC able to handle non-spherical particles in general? Can you speculate about the retrieval errors caused by assuming spherical particles for non-spherical ones?

The MISR results are available at https://appliedsciences.nasa.gov/our- impact/news/misr-plume-heights-and-aerosol-characteristics-2019-raikoke-eruption, although the linked Powerpoint file seems corrupted at the moment.

ORAC can be run on non-spherical particles. Essentially the LUT part of the code (DISORT) would be run using T-matrix theory. However, we assumed spherical particles for two main reasons: (1) At thermal infrared wavelengths larger than 10 µm the particle habit (non-sphericity) is expected to have little impact on the retrievals as found by numerous previous authors (Wen and Rose, 1994; Corradini et al., 2008; Clarisse et al., 2010; Newman et al., 2012; Pavolonis et al., 2013). Yang et al. (2007) present a study that looked at this exact issue but for desert dust (that is similar in many ways to volcanic ash) and showed that errors in assuming non-spherical particles at thermal infrared wavelengths are negligible (note though that this is not the case at the VIS/NIR wavelengths that MISR measures). (2) The exact non-spherical shapes of the Raikoke ash particles under investigation here are unknown. Therefore, approximating their shape by some other irregular shape may introduce further error than simply assuming a sphere. We recognise that it's possible to find differences between spherical and non-spherical particles if irregular, porous objects are compared with spheres. Kylling et al. (2014) found that differences in the total mass uncertainty would increase from 40% to 50%. However, it is questionable how representative the particle shapes used in the Kylling study are for the Raikoke ash and therefore we cannot conclude that the 10% uncertainty found by Kylling et al. (2014) would apply here. We have added this discussion to the revised manuscript.

Lines 285-286 and Eq. 10: The definition of ash mass loading is indeed the same as that of cloud liquid water path (LWP) and assumes no vertical variation in cloud/ash particle size. For marine stratocumulus clouds, however, effective radius often increases linearly from cloud base to top. Because the effective radius derived from VIS-NIR bispectral observations is representative of the cloud top, the assumed vertical homogeneity results in an overestimated LWP. For such boundary layer clouds, an 'adiabatic' model has been proposed, which leads to a proportionality factor of 10/9 instead of 4/3 in Eq. 10. This adiabatic correction reduces the VIS-NIR LWP by 17%, which agrees better with microwave LWP estimates for marine Sc.

Your estimate of the Raikoke ash cloud geometric thickness is 1.04±0.56 km, which is larger than the $\sim$300 m typical thickness of marine Sc. Thus, vertical variations in ash particle effective radius can also introduce biases in ash mass loading when vertical homogeneity is assumed.

Can you comment on which part of the ash cloud (top, middle, bottom) your IR-based effective radius estimates are representative of? Do you expect vertical variations in effective radius—perhaps a decrease from base to top, in contrast to marine Sc—and how would these bias your ash mass load estimates?

The effective radius retrieved from thermal infrared channels will be representative of the effective emission height of the volcanic cloud. This is often found to be near the middle of the volcanic ash cloud. For marine Sc clouds there is a good physical justification for the 10/9 factor (liquid water droplet growth with the lapse rate). However, for volcanic ash clouds the change in particle size with vertical depth is highly uncertain. There are physical arguments that one could follow in that there may be some sorting going on - large particles falling out first and therefore small-to-large particle gradient from cloud-top to -base might be expected at the beginning of the eruption due to gravitational settling. However, as drifting ash clouds disperse it's possible that the remaining small ash particles are uniformly distributed in the vertical (meaning that the 4/3 is more representative). This discussion also does not take into account more complex processes that are known to occur in ash clouds such as ash aggregation and wet deposition. A recent paper by Saxby et al. (2019) shows that $\sim$1 micron particle sizes can maintain heights anywhere from 8–14 km over $\sim$6 days which suggests that particle fall velocities for this size are strongly influenced by advection and turbulent diffusion. In lieu of any clear evidence (observations) of a vertical gradient in particle size that would be representative of ash particle dispersion over the first week, we decided to use the first-order assumption of a uniform vertical distribution.

Line 344: "there is no information on particle size for opaque plumes". Perhaps you could note this limitation earlier, when describing the method in section 3.

Done.

Lines 359-360: You state that "The ORAC heights are, however, an improvement to simply matching a brightness temperature to a NWP profile…"; but, you don't demonstrate this. Could you perhaps plot the height or height range derived by matching the minimum plume BT within the black circle in Fig. 3a to the ERA5 profile?

Thank you for raising this point. Addressing this comment led us to review the CTH time-series derived from ORAC data. Upon careful inspection, we found that using the 15 km radius was actually leading to overestimation of the duration of each eruptive pulse. Essentially, by having the search radius too large the maximum CTH corresponds to parts of the plume after it has detached from the volcano, leading to a time-series that overestimates the duration of each eruptive episode. To resolve this, we varied the search radius until an optimal radius was found. We define 'optimal' as the search radius that resulted in the closest match (minimised sum of squared differences) to the GOES-17 (G17) sideview height data. This approach solves the issue of overestimation of eruptive episodes but also ensures a better comparison to G17 as we do not know exactly which AHI pixel corresponds to the G17 sideview height (latitude/longitude coordinates are not provided in Supp. Matt. of Horváth et al. (2021)). The optimal search radius is 7.5 km and this is what we present in our revised submission.

As suggested, we have also compared the ORAC heights to the minimum brightness temperature-NWP profile matching method (hereafter 'BTmin method'). To make this comparison we used the bounding box used by McKee et al. (2021) to re-create their CTH timeseries (note though that McKee et al. use ERA-Interim whereas we use ERA5). What we found was that the two methods give quite similar results when compared to the G17 heights (similar correlation coefficient and precision). However, the largest difference was the bias. The BTmin method has a bias of -1.92 km, while ORAC's bias compared to G17 is -0.75 km (-0.32 km if overshoots are omitted). This result is to be expected because the BTmin method will, by definition, never produce heights in the stratosphere. Further, while ORAC effectively matches the height to the temperature profile, it accounts for other factors (cloud emissivity/transmission, viewing angle, above cloud transmission) compared to matching the raw BT. We have updated Fig. 3b showing the time-series of the two methods (in addition to the lapse rate tropopause) in our revision. The updated figure is shown here as Fig.4.

Lines 360-362: How big are the differences in the 5 cost values? Are these differences significant?

The differences in the 5 cost values vary over several orders of magnitude in some parts of the ash cloud but can be quite similar (same order of magnitude) in other parts. The most notable differences in cost amongst the 5 retrieval setups are seen when comparing the stratospheric *a priori* retrieval configurations to the tropospheric ones. To illustrate the relative differences in the 5 cost values, we have generated a new figure (added to the revised manuscript) that shows each cost map, the combined cost map (where the minimum cost for each configuration is selected on a per-pixel basis) and a 'classification map' where we have coloured each pixel according to the retrieval configuration that resulted in the lowest cost. We have also plotted the $T_{11}$ brightness temperature and a natural-colour composite to provide contextual information that shows that the selected forward model configuration is reasonable (e.g. the stratospheric ash over 500 hPa water cloud is selected for ash overlying the cold cloud associated with the cyclone). We have added text to discuss this new figure (and

[Figure]

Figure 4: Updated Fig. 3 of original manuscript.

analysis) in our revision.

Lines 389-391: You suggest that the increase in effective radius is due to a retrieval artefact at large satellite view zenith angle (VZA). Cloud drop effective radius retrieved by the plane-parallel VIS-NIR bispectral method has been found to increase at large VZA due to 3D effects (shadowing, illumination) affecting the VIS reflectances (Maddux et al., 2010, 10.1175/2010JTECHA1432.1; Grosvenor and Wood, 2014, 10.5194/acpd-14-303- 2014; Horvath et al. 2014, 10.1002/2013JD021355).

How does VZA affect your IR measurements that could explain this retrieval artefact? Is it differential limb cooling in the various channels due to increased absorption by $CO_2$ and/or water vapor? Can you explain it by the topology (shape of constant effective radius curves) of the ORAC look-up tables plotted in Fig. 10?

The forward model was not written to cope with high zenith angles and so the assumptions (e.g. nadir-viewing satellite, plane-parallel ash cloud etc) at extreme view angles all breakdown. The LUTs shown in Fig. 10 become distorted at high view angles and so cannot be interpolated reliably, resulting in retrieval artefacts. The answer as to why we see an increase in the retrieved effective radius is not a simple one. Certainly limb darkening could be having an effect, in addition to pixel distortion due to extreme viewing geometry and parts of the ash cloud exiting the field-of-view of Himawari. To fully appreciate the impact, more advanced radiative transfer simulations (accounting for Earth's curvature and limb-viewing geometry) would be required and is beyond the scope of the

[Figure]

Figure 5: Updated Fig. 7 of original manuscript.

present study. We do note that Gu et al. (2005) noticed the same issue with their analysis - GOES retrievals at zenith angles of ∼70 degrees resulted in an overestimation of effective radius when compared with close-to-nadir (MODIS and AVHRR) retrievals. In our revision, we have added reference to the Gu et al. (2005) paper and have shaded the region in the time-series where the ORAC retrievals may be impacted by this issue.

Lines 459-460: You state that there is generally good agreement between the ORAC and validation heights. However, Fig. 7 suggests that the agreement between ORAC and GOES-17 is better in the troposphere, while the skill of ORAC in the stratosphere seems limited. GOES-17 indicates heights between 10-16km, while ORAC retrieves near constant heights around 12-13km. You state elsewhere that the IR measurements had little influence on the retrieved stratospheric heights, which were close to the chosen a priori near 12km. Perhaps a more nuanced statement on the height comparison is needed here.

The reviewer has correctly interpreted our results; however, to clarify our findings further, we have split the height validation into two parts: near-source plume (comparison with GOES-17) and distal plume (comparison with CALIPSO). The updated Fig. 7 is included here as Fig. 5. What we find is that the ORAC heights show very good agreement with the G17 heights (bias=-0.75 km, precision=1.78 km, R=0.84) for the near-source plume, but for the distal ash, there is a notable negative bias (-2.67 km) when compared to several CALIOP passes. As suggested, in our revision, we have re-iterated that ORAC showed limited skill in retrieving heights in the stratosphere.

Lines 479-482: Purely as a suggestion for future study, I recommend analysing ORAC retrievals for the 2021 La Soufriere eruptions, where the tropical temperature profile shows a sharp inversion at the tropopause and distinct temperature increase in the stratosphere. ORAC stratospheric height retrievals might have better skill in a tropical case than in the current mid-latitude case characterized by an isothermal lower stratosphere. CALIOP and GOES-17 height estimates are available for La Soufriere too.

Thank you suggesting this. Our group is currently working on this case study and indeed we agree that the atmospheric conditions for La Soufriere should lead to better constrained results for height retrievals in the stratosphere.

Lines 529-534: Bruckert et al. (2022, 10.5194/acp-22-3535-2022) in the same special issue also show that resolving the individual phases of the 2019 Raikoke eruptions improves the total ash burden forecast and reduces ash mass overestimation, compared to using continuous and constant eruption source parameters.

Thanks. We've added reference to this paper in our revised submission.

**Technical corrections**

Line 49: "in satellite-based, ash cloud retrieval" -> remove the comma

Done.

Line 173: "interpolaRted" -> "interpolated"

Done.

Line 183: Is $\lambda$ the wavelength?

Yes. Clarified.

Line 258: "problem of a multiple solutions" -> remove "a"

Done.

Line 265: "a tropospheric, a priori ash layer pressure" -> remove the comma

Done.

Line 268: "which is a typical of value for ash" -> remove "of"

Done.

Line 417: "side-view, times-series" -> remove the comma

Done.

Line 449: "were  0.3–0.7 km according to" -> "…were  0.3–0.7 km thick…"

Done.

Line 455: "would be fall below" -> remove "be" or "fall"

Done.

Line 501: "their is very little" -> "there"

Done.

Fig. 8a: I presume, the black line is the CALIOP ground track with green indicating the section plotted in panel c.

Yes it is. Clarified in the figure caption.

Line 504: "for a cloud layers" -> "layer"

Done.

Line 602: "converts to a different in absolute" -> "difference"

Done.

**Additional technical corrections**

- The following statement in the abstract was poorly worded:
  "We also find that our implementation of the ORAC algorithm was reliable out to four days and was able to track the median ash cloud at concentrations below peak ash concentration safety limits (< 4 mg m-3) if typical ash cloud geometric thicknesses were assumed."
  We have re-phrased the statement regarding concentration safety limits so that it explicitly refers to Fig. 8 of the revised manuscript (Fig. 6 of the original manuscript). The revised statement is:
  "We also find that median ash cloud concentrations fall below peak ash concentration safety limits ($<$4 mg m$^{-3}$) 11–16 h after the eruption begins, if typical ash cloud geometric thicknesses are assumed."

- Typo in Eq. (5). This has been corrected by replacing $\frac{\partial L_{bc,a}^{\uparrow}}{\partial T_{s,a}}$ with $\frac{\partial B(T_{s,a})}{\partial T_{s,a}}$.

- Fig. 9 of original manuscript has been updated to plot data on higher resolution vertical grid. i.e. 137 levels for transmittance, temperature and specific humidity profiles, 68 levels for weighting function (i.e. derivative of transmittance with respect to ln(pressure)). Previously all data was plotted on a coarse grid (51 levels).

**References**

L. Clarisse, D. Hurtmans, A. J. Prata, F. Karagulian, C. Clerbaux, M. De Mazière, and P.-F. Coheur. Retrieving radius, concentration, optical depth, and mass of different types of aerosols from high-resolution infrared nadir spectra. *Applied Optics*, 49(19):3713, July 2010. ISSN 0003-6935, 1539-4522. doi: 10.1364/AO.49.003713. URL `https://www.osapublishing.org/abstract.cfm?URI=ao-49-19-3713`.

S. Corradini, C. Spinetti, E. Carboni, C. Tirelli, M. Buongiorno, S. Pug-
    naghi, and G. Gangale. Mt. Etna tropospheric ash retrieval and
    sensitivity analysis using moderate resolution imaging spectrora-
    diometer measurements. *Journal of Applied Remote Sensing*, 2(1):
    023550, Nov. 2008. ISSN 1931-3195. doi: 10.1117/1.3046674. URL
    `http://remotesensing.spiedigitallibrary.org/article.aspx?doi=10.1117/1.3046674`.

Y. Gu, W. I. Rose, D. J. Schneider, G. J. S. Bluth, and I. M. Watson.
    Advantageous GOES IR results for ash mapping at high latitudes:
    Cleveland eruptions 2001. *Geophysical Research Letters*, 32(2),
    2005. ISSN 0094-8276. doi: 10.1029/2004GL021651. URL
    `http://doi.wiley.com/10.1029/2004GL021651`.

N. J. Harvey, H. F. Dacre, C. Saint, A. T. Prata, H. N. Webster, and R. G.
    Grainger. Quantifying the impact of meteorological uncertainty on emis-
    sion estimates and the risk to aviation using source inversion for the Raikoke
    2019 eruption. *Atmospheric Chemistry and Physics*, 22(13):8529–8545,
    July 2022. ISSN 1680-7324. doi: 10.5194/acp-22-8529-2022. URL
    `https://acp.copernicus.org/articles/22/8529/2022/`.

□. Horváth, O. A. Girina, J. L. Carr, D. L. Wu, A. A. Bril, A. A. Mazurov, D. V.
    Melnikov, G. A. Hoshyaripour, and S. A. Buehler. Geometric estimation of
    volcanic eruption column height from GOES-R near-limb imagery – Part 2:
    Case studies. *Atmospheric Chemistry and Physics*, 21(16):12207–12226,
    Aug. 2021. ISSN 1680-7324. doi: 10.5194/acp-21-12207-2021. URL
    `https://acp.copernicus.org/articles/21/12207/2021/`.

A. Kylling, M. Kahnert, H. Lindqvist, and T. Nousiainen. Volcanic ash infrared
    signature: porous non-spherical ash particle shapes compared to homoge-
    neous spherical ash particles. *Atmospheric Measurement Techniques*, 7(4):
    919–929, Apr. 2014. ISSN 1867-8548. doi: 10.5194/amt-7-919-2014. URL
    `https://www.atmos-meas-tech.net/7/919/2014/`.

G. R. McGarragh, G. E. Thomas, A. C. Povey, C. A. Poulsen, and R. G.
    Grainger. Volcanic ash retrievals using ORAC and satellite measurements
    in the visible and IR. *Proc. 'ATMOS 2015, Advances in Atmospheric Science
    and Applications', Heraklion, Greece, 8–12 June 2015 (ESA SP-735, Novem-
    ber 2015)*, page 8, 2015.

K. McKee, C. M. Smith, K. Reath, E. Snee, S. Maher, R. S. Matoza,
    S. Carn, L. Mastin, K. Anderson, D. Damby, D. C. Roman, A. Degterev,
    A. Rybin, M. Chibisova, J. D. Assink, R. de Negri Leiva, and A. Perttu.
    Evaluating the state-of-the-art in remote volcanic eruption character-
    ization Part I: Raikoke volcano, Kuril Islands. *Journal of Volcanol-
    ogy and Geothermal Research*, 419:107354, Nov. 2021. ISSN
    03770273. doi: 10.1016/j.jvolgeores.2021.107354. URL
    `https://linkinghub.elsevier.com/retrieve/pii/S0377027321001839`.

S. M. Newman, L. Clarisse, D. Hurtmans, F. Marenco, B. Johnson, K. Turnbull,
    S. Havemann, A. J. Baran, and J. Haywood. A case study of observations of
    volcanic ash from the Eyjafjallajökull eruption: 2. Airborne and satellite radia-
    tive measurements. page 19, 2012.

M. J. Pavolonis, A. K. Heidinger, and J. Sieglaff. Automated retrievals of volcanic ash and dust cloud properties from upwelling infrared measurements: Retrieval of ash/dust cloud properties. *Journal of Geophysical Research: Atmospheres*, 118(3):1436–1458, Feb. 2013. ISSN 2169897X. doi: 10.1002/jgrd.50173. URL `http://doi.wiley.com/10.1002/jgrd.50173`.

C. A. Poulsen, R. Siddans, G. E. Thomas, A. M. Sayer, R. G. Grainger, E. Campmany, S. M. Dean, C. Arnold, and P. D. Watts. Cloud retrievals from satellite data using optimal estimation: evaluation and application to ATSR. *Atmospheric Measurement Techniques*, 5(8):1889–1910, Aug. 2012. ISSN 1867-8548. doi: 10.5194/amt-5-1889-2012. URL `https://www.atmos-meas-tech.net/5/1889/2012/`.

F. Prata and I. Grant. Determination of mass loadings and plume heights of volcanic ash clouds from satellite data, 2001. URL `http://rgdoi.net/10.13140/RG.2.1.4405.2327`.

G. S. Prata, L. J. Ventress, E. Carboni, T. A. Mather, R. G. Grainger, and D. M. Pyle. A New Parameterization of Volcanic Ash Complex Refractive Index Based on NBO/T and $SiO_2$ Content. *Journal of Geophysical Research: Atmospheres*, 124(3):1779–1797, Feb. 2019. ISSN 2169897X. doi: 10.1029/2018JD028679. URL `http://doi.wiley.com/10.1029/2018JD028679`.

J. Saxby, A. Rust, K. Cashman, and F. Beckett. The importance of grain size and shape in controlling the dispersion of the Vedde cryptotephra. *Journal of Quaternary Science*, Nov. 2019. ISSN 0267-8179, 1099-1417. doi: 10.1002/jqs.3152. URL `https://onlinelibrary.wiley.com/doi/abs/10.1002/jqs.3152`.

S. Smirnov, I. Nizametdinov, T. Timina, A. Kotov, V. Sekisova, D. Kuzmin, E. Kalacheva, V. Rashidov, A. Rybin, A. Lavrenchuk, A. Degterev, I. Maksimovich, and A. Abersteiner. High explosivity of the June 21, 2019 eruption of Raikoke volcano (Central Kuril Islands); mineralogical and petrological constraints on the pyroclastic materials. *Journal of Volcanology and Geothermal Research*, 418:107346, Oct. 2021. ISSN 0377-0273. doi: 10.1016/j.jvolgeores.2021.107346. URL `https://www.sciencedirect.com/science/article/pii/S037702732100175X`.

P. Watts, C. Mutlow, A. Baran, and A. Zavody. Study on cloud properties derived from Meteosat Second Generation observations. *EUMETSAT ITT*, 97:181, 1998.

S. Wen and W. I. Rose. Retrieval of sizes and total masses of particles in volcanic clouds using AVHRR bands 4 and 5. *Journal of Geophysical Research*, 99(D3):5421, 1994. ISSN 0148-0227. doi: 10.1029/93JD03340. URL `http://doi.wiley.com/10.1029/93JD03340`.

C. Witham, M. Hort, D. Thomson, B. Devenish, H. Webster, and F. Beckett. The current volcanic ash modelling set- up at the London VAAC. page 11, 2019.

K. Yang, N. A. Krotkov, A. J. Krueger, S. A. Carn, P. K. Bhartia, and P. F. Levelt. Retrieval of large volcanic $SO_2$ columns from the Aura Ozone Monitoring

Instrument: Comparison and limitations. *Journal of Geophysical Research*, 112(D24), Nov. 2007. ISSN 0148-0227. doi: 10.1029/2007JD008825. URL `http://doi.wiley.com/10.1029/2007JD008825`.